# Zoning Strategy for Basin Land Use Optimization for Reducing Nitrogen and Phosphorus Pollution in Guizhou Karst Watershed

Xu Zhou [1], Wenbin Zhang [1], Yu Pei [1], Xiao Jiang [1] and Shengtian Yang [1,2,*]

1 School of Geography and Environmental Sciences, Guizhou Normal University, Guiyang 550025, China
2 College of Water Sciences, Beijing Normal University, Beijing 100875, China
* Correspondence: yangshengtian@bnu.edu.cn

**Abstract:** Eutrophication caused by excessive total nitrogen (TN) and total phosphorus (TP) emissions is of wide concern for society at large. Studies have revealed certain relationships among land use, TN, and TP. However, the relationships among land use compound topographic position, TP, and TN have seldom been studied. Therefore, the objectives of this paper are to construct optimal zoning of land use and reduce the nutrient load of lakes. Spearman correlation and redundancy analyses were used to reveal the relationship between land use comprehensive topographic position and TN and TP in the lakes of Guizhou Plateau. The results show that the nutritional state of the research area is medium. The trophic level index (TLI) value and TN concentration were high during flood periods, while TP concentration was high in dry periods. The TN concentration in the tributaries was higher than that in the reservoir area. Construction land and valley were the sources of the pollution, whereas forest land and gentle slope were the sink. According to the "source–sink" effect, once the optimal zoning of land use is completed, the governance of urban land pollution governed areas should be strengthened next. This paper can provide decision support for water environment management and sustainable development decision-making.

**Keywords:** total nitrogen; total phosphorous; land use; topographic position; pollution source control zoning



## 1. Introduction

Population growth and social economic development have led to severe water eutrophication due to agricultural pollution, threatening global ecological security, water source health, sustainable economic development, and social harmony and stability [1,2]. In China, agricultural activities have become the primary source of water pollution throughout China [3]. Pollutants (such as sediment, nutrients, and pesticides) resulting from agricultural and human activities enter water and soil and are discharged via surface runoff or leaching [2], thereby posing a serious threat to the environment. This has become a domestic and global environmental problem to be solved urgently.

Nitrogen and phosphorus are important in food production and for all life on Earth [4], but excessive nitrogen and phosphorus loads pose major threats to the aquatic ecosystems [5]. The Water Research Council of South Africa reports that 54% of rivers and lakes in Asia are facing eutrophication problems [6]. Certain countries implemented measures for the prevention and control of eutrophication caused by non-point source pollution. Currently, water pollution in river basin has been studied globally. For example, Japan has formulated a water quality protection plan to control water pollution in the Biwa Lake Basin [7]. In 1978, the United States cleaned up the bottom mud of Lilly Lake to control water pollution [8]. However, the eutrophication of rivers and lakes in China remains problematic [9]. As per the Second National Survey of Pollution Sources released by China, the total nitrogen (TN) and total phosphorus (TP) loads in lakes and rivers in

China accounted for 3,041,400 and 315,400 tons, respectively, in 2020 [10]. The 2019 Chinese Lake Survey reported high nutrient levels in the lakes in the eastern Chinese plain and the Yunnan–Guizhou Plateau Lake region [11]. Studies by Han et al. [12] and Xu et al. [13] demonstrated that nutrients in lakes are primarily supplied by tributaries and nutrient concentrations in tributaries are higher than in reservoirs. Therefore, by optimizing the division of land functions over the whole basin, we could slow the pollution entering in the tributaries. This approach is expected to be more scientifically reasonable compared to controlling pollution in the lake and reservoir area.

Land use affects ecological processes such as the basin water cycle and material migration [14]. Usually, agricultural land and construction are associated with high nutrient concentration in rivers [15], whereas forest land can intercept and filter pollutants in water bodies [16]. The value of land use to water management is understood well and has been examined to some extent. For instance, Bahman and Kaniyuke [17] examined the correlation between land use and various indicators of river water quality in the Chugoku area of Japan. They demonstrated that 92% and 64% of the concentration changes in dissolved oxygen and TP, respectively, were attributed to change of land use. Zampella et al. [18] reported a significant correlation between land use patterns and water quality indicators on different spatial scales in Pinelands Basin, New Jersey. They demonstrated that the water quality will be affected if the land use change rate in the basin exceeds 10%. However, the relationship between land use composition and nutrient concentration presents considerable regional differences reflecting the variable water morphologies, hydrologies, geographies, and human activities in different areas [19–22]. Therefore, based on land use, this study comprehensively considers the influences of topographic positions on the watershed water quality. In addition, to strengthen the research contents, this study comprehensively considered the effects of different land use types on different slope position ("L-SP", land use-slope position) on watershed water quality. The aim is to understand and develop schemes for pollution control and environmental treatment of river basin water.

The Guizhou Plateau Lake, located in the Karst region, has a shallow soil layer and a weak water-holding capacity leading to a number of environmental problems such as frequent shortage of surface water, fragile ecology, and rapid diffusion of pollutants [23]. Comparing the water environment conditions in typical lakes in China, researchers reported that nitrogen and phosphorus are far more concentrated in Guizhou Plateau Lake than in other lakes [24–26]. In recent years, the research data from Guiyang environmental monitoring station and "two lakes one reservoir" management monitoring center station showed that the trophic level index (TLI) in Hongfeng Lake was in the range of 36–53 (typically > 40), indicating a medium eutrophication state [27]. Furthermore, Baihua Lake is eutrophic during the flood season and mesotrophic during the dry season, thus indicating a pessimistic water quality of reservoirs in the drinking-water source area of Guizhou Province [22]. Most studies on eutrophication in China's plateau lake basin have analyzed only the relationship between land use and water quality. The reduction of non-point source pollution in the plateau lake basin has not been systematically analyzed. Only Dai et al. [28] attempted to relate land use and water quality to differential pollution control. By quantitatively assessing the impact of land use on water quality in the plateau lake basin, we could establish control zones suitable for plateau lake basins in China and thereby reduce the entry of exogenous nutrients in the lake and non-point source pollution of the lake. Such assessments would provide government agencies and decision makers with improved lake control programs and restoration plans. Therefore, they have important reference value and practical significance for plateau lake basin management.

This study analyzes the pollution situation in the HBA watershed, a typical lake in Karst region. The aims of this study are threefold: (1) identifying the spatial and temporal distribution characteristics of TN and TP in the study area and calculating the nutrient statuses of the basin over the years; (2) discussing the effects of topographic-slope compound land use structure on nitrogen and phosphorus in the basin; and

(3) establishing pollution control zoning of different regions in the HBA watershed. Objective (1) is achieved by using the comprehensive nutrient index method and comparing the TN and TP concentrations in the reservoir area and tributaries; objective (2) is attained by combining statistical methods, remote sensing, and geographic information systems technology; and objective (3) is accomplished based on the results of previous analyses.

## 2. Materials and Methods

### 2.1. Study Area

The research area includes the Hongfeng Lake Basin, Baihua Lake Basin and Aha Reservoir Basin, collectively referred to as the HBA watershed. The area is located in the middle of Guizhou Karst Plateau in southwest China. It occupies the water–land connecting zone between Guiyang City (the provincial capital) and Gui'an New Area, and the intersection zone between the built area and farming area (26°8′–26°43′ E, 105°58′–106°40′). The HBA watershed has obvious plateau and mountainous features. The terrain is high in the southwest and low in the northeast, with highest and lowest elevations of 1728.20 and 1070.02 m, respectively. The total basin area of the HBA watershed (2085.50 km$^2$) includes the Hongfeng Lake Basin area (1596.00 km$^2$), the Baihua Lake Basin area (299.00 km$^2$), and the basin area of the Aha Reservoir (190.05 km$^2$). The HBA watershed is a typical rain source basin in a plateau mountainous area with a runoff mostly supplied by precipitation. The annual variation of rainfall in the basin is small but varies in spatial and temporal distribution. The annual average rainfall was 1129.5 mm, of which 593 mm in dry season and 1386 in wet season. Temporally, the rainfall is abundant in the flood season from May to October and low in the dry seasons from January to April and October to December.

Most of the Gui'an New Area (75.22%) is located in the upper reaches of the basin. The rapid transformation of land types in the historical period and current high-intensity urbanization construction and agricultural development activities in Gui'an New Area have severely threatened the water security of the basin. The HBA watershed provides three important drinking water sources to Guiyang City. According to the "Surface water environmental quality standard (GB3838-2002)," surface waters in China are divided into five categories, of which one to five categories represent the water quality from the best to the worst. In the 1980 s, the water quality was basically stable in Category II. During the flood season of 1995, "black water" periodically appeared in Hongfeng Lake. The water environment management department continues to perform water eutrophication prevention and control work with certain amount of success, but the coal mining problem at the Aha Reservoir remains as a hidden danger of eutrophication events. Therefore, the HBA watershed urgently requires additional reasonable and effective land use control strategies.

### 2.2. Data

#### 2.2.1. Water Quality Sampling

The water quality monitoring points are distributed in terms of functions of the drinking water source, geomorphic and hydrological conditions, basin areas, and other features. On the basis of the sub-basin division, 14 main tributaries in the HBA watershed were selected. The water quality monitoring points were the inlets to each tributary of the sub-basin: Hongfeng Lake Basin (4 points), Baihua Lake Basin (5 points), and the Aha Reservoir Basin (5 points). Moreover, according to the water source protection division, 14 water quality monitoring points were set up in the water area of the HBA Watershed: the Hongfeng Lake Basin (7 points), Baihua Lake Basin (5 points), and Aha Reservoir Basin (2 points). Thus, a total of 28 monitoring points were arranged. In Hongfeng Lake and Baihua Lake, water quality was sampled on a monthly basis from 2013 to 2018. In the Aha Reservoir, water quality was sampled at three-monthly intervals from 2013 to 2014 and on a monthly basis from 2015 to 2018. The eutrophic status of the water source area in HBA watershed was evaluated by multiple water quality indices: TP, TN, permanganese index (CODMn), transparency (SD), and chlorophyll (Chla) in the reservoir area and TP and TN in the tributaries. The water quality sampling and water quality indices strictly

complied with the standard methods of China's Water and Wastewater Monitoring Methods (Fourth Edition).

### 2.2.2. Division of Monitoring Unit

The watershed division method was based on the research of Levesque et al. [29]. Division into sub-basins was based on data from the digital elevation model on the Geospatial Data Cloud (http://www.gscloud.cn/, accessed on 13 January 2019). The basin was segmented by ArcSWAT watershed extraction module as per the flow direction of the river system, the intersection points of trunks and tributaries, and the outlet. After merging the sub-basins as per the characteristics of the primary tributaries in the basin of two lakes and one reservoir, 14 sub-basins were finally generated, each corresponding to a water quality monitoring point. The first- and second-level protection zones and quasi-protection zones of drinking water source in the HBA watershed formed separate monitoring units with monitoring points in their scope. After adding seven monitoring units in the protection zone of the drinking water source, the total number of monitoring units was 21 (Figure 1 and Table 1).

| Basic situation of the study area | Latitude and longitude | Elevation Range | Rainfall |
|---|---|---|---|
| | It is located at 105 ° 58′ -106 ° 40′ N, 26 ° 8′ -26 ° 43′ E, the Central Province of Guiyang, next to the provincial capital. | 1070.02m$^{-1}$ 728.20m | Most of the runoff comes from precipitation. The interannual variation of rainfall is small and the spatial and temporal distribution is uneven. The annual average rainfall was 1129.5 mm, of which 593 mm in dry season and 1386 in flood season. The precipitation is mainly concentrated in summer and autumn, especially in may-august. |
| **Basic situation of the study area** | **Temperature** | **Terrain** | **Landform** |
| | The average temperature is about 15 degrees. In July, the hottest month, the average temperature is 24 degrees. In January, the coldest month, the average temperature is 4.6 degrees. The study area is a central Asian tropical monsoon climate with no heat in summer and no cold in winter. | The terrain undulates greatly. It is high in the southwest and low in the northeast | In the karst area, the karst are well developed, the Peak Forest Basin and the Peak Forest Valley have prominent morphological characteristics, and the others have mountains, falling water caves, dry gullies and so on. |

**Figure 1.** *Cont.*

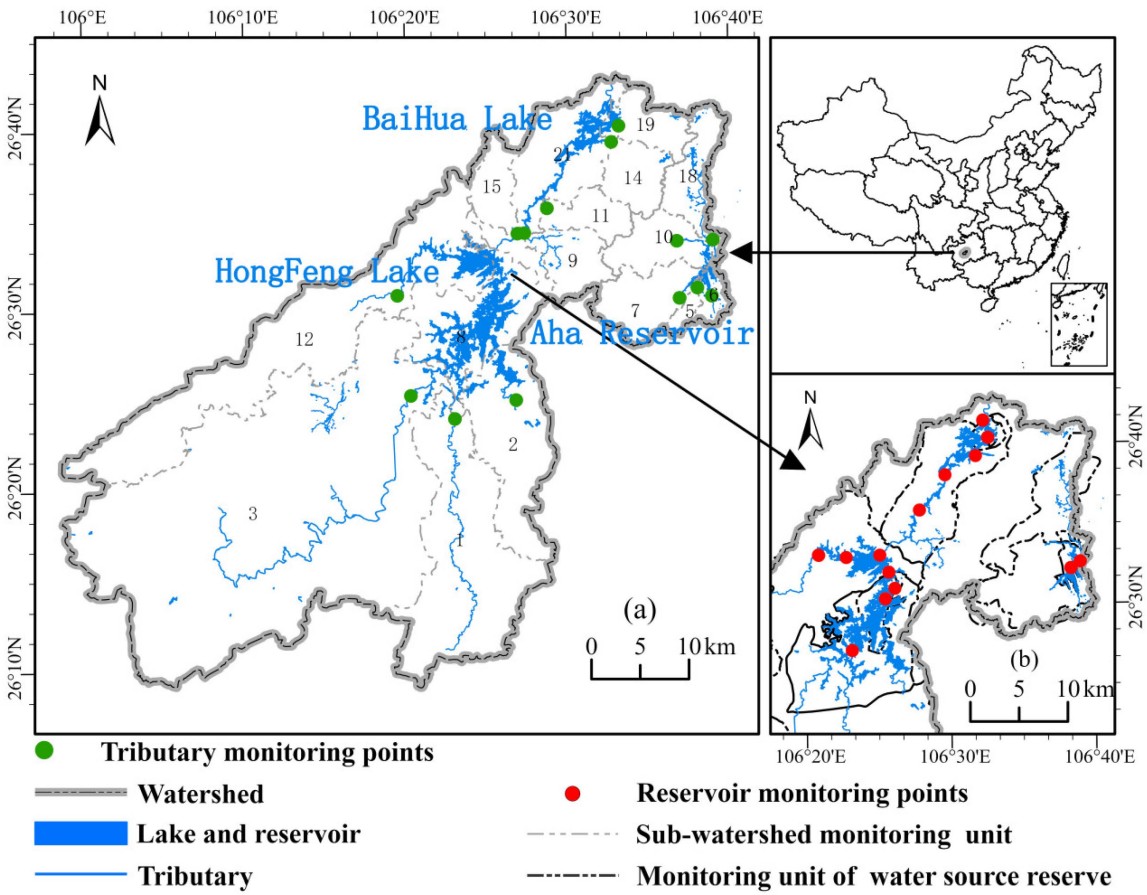

**Figure 1.** Locations and monitoring points in the study area: (**a**) Tributary monitoring points; (**b**) Reservoir monitoring points.

**Table 1.** Monitoring units and their corresponding monitoring points.

| ID | Sample Point Name | Watershed | Point Code | Longitude | Latitude |
|----|-------------------|-----------|------------|-----------|----------|
| 1 | MaiWeng river | | T1 | 106°19′24″ | 26°30′51″ |
| 2 | YangCang river | | T2 | 106°20′11″ | 26°25′15″ |
| 3 | HouLiu river | | T3 | 106°26′40″ | 26°24′59″ |
| 4 | MaXian river | | T4 | 106°22′53″ | 26°24′00″ |
| 5 | HouWu | | L1 | 106°25′30″ | 26°30′21″ |
|  | XiJiao | Hongfeng Lake | L2 | 106°26′9″ | 26°31′1″ |
| 6 | HuaYuDong | | L3 | 106°25′45″ | 26°32′4″ |
|  | Sancha River | | L4 | 106°23′11″ | 26°27′8″ |
|  | DaBa | | L5 | 106°25′8″ | 26°33′5″ |
| 7 | YaoDng | | L6 | 106°22′48″ | 26°32′58″ |
|  | PianShanZhai | | L7 | 106°20′54″ | 26°33′4″ |
| 8 | LiJiachong river | | T5 | 106°32′43″ | 26°39′16″ |
| 9 | SongJiachong river | | T6 | 106°33′9″ | 26°40′9″ |
| 10 | DongMenQiao river | | T7 | 106°27′18″ | 26°34′15″ |
| 11 | GaoJia river | | T8 | 106°28′42″ | 26°35′37″ |
| 12 | ChangChong river | | T9 | 106°26′52″ | 26°34′13″ |
| 13 | GuiLv | Baihua Lake | L8 | 106°32′43″ | 26°40′21″ |
| 14 | BaiDaBa | | L9 | 106°32′22″ | 26°38′52″ |

**Table 1.** *Cont.*

| ID | Sample Point Name | Watershed | Point Code | Longitude | Latitude |
|---|---|---|---|---|---|
| | MaiXi | | L10 | 106°31′50″ | 26°39′13″ |
| 15 | YanJiaozhai | | L11 | 106°29′42″ | 26°38′03″ |
| | Hua Qiao | | L12 | 106°27′55″ | 26°35′53″ |
| 16 | LanNiGou | | T10 | 106°38′49″ | 26°30′38″ |
| 17 | CaiChongGou | | T11 | 106°37′56″ | 26°31′6″ |
| 18 | YouYu river | | T12 | 106°36′50″ | 26°30′33″ |
| 19 | BaiYan river | Aha Reservoir | T13 | 106°36′42″ | 26°33′43″ |
| 20 | JinZhong river | | T14 | 106°38′56″ | 26°33′46″ |
| 21 | Reservoir | | L13 | 106°38′21″ | 26°32′10″ |
| | Reservior eastern | | L14 | 106°38′21″ | 26°32′11″ |

T: tributary; L: reservoir.

### 2.2.3. Land Use Data

The data were high-quality Gaofen-1 satellite data with cloud cover of <10% that was collected in 2013 and 2016. The data were downloaded from the website of the China Resources Satellite Data and Application Center (http://www.cresda.com/) on 10 January 2020. In preparation for land use type extraction, the Gaofen-1 images were subjected to radiation calibration and Flaash atmospheric correction using ENVI 5.3 tools. Land use types were classified using maximum likelihood and visual interpretation. Field surveys were performed from 21 May to 23 May 2016, and visits to understand the land cover were made in 2013. Through these surveys, ~400 actual land use type verification points were identified on the ground. In areas that are difficult for human footprints to reach, the results were confirmed from unmanned aerial vehicle images and Google Earth images. The high Kappa coefficient (0.9576) confirmed that the classification accuracy meets the research requirements. Finally, the land use types were divided into woodland, dry farmland, paddy land, grassland, water body, unused land, and construction land. Moreover, according to the territorial spatial planning and functions of land use types, the forest land, grassland, and water bodies were classified as ecological land; dry farmland and paddy land were classified as agricultural land; and construction land was classified as urban land.

### 2.2.4. Topographic Position Index

The topographic position index (TPI) was generated using Jenness' extension in ArcGIS 10.5. The TPI is classified by landscape and geomorphic type [30]. Landscape is divided into three categories according to slope position: up, medium and down. Geomorphologic types include plains, valleys, and ridges. In this study, the TPI was divided into river valleys, gentle slopes, steep slopes, and ridges. As the TPI is scale-dependent, it was evaluated on local (200 m) and regional (500 m) scales. The standardized TPI is formulated as follows:

$$standardized\text{TPI} = \frac{\text{TPI} - \text{TPI}_{mean}}{\text{TPI}_{standard\ deviation}} \tag{1}$$

### 2.3. Methods

### 2.3.1. Methods for Evaluating Changes in Nutrient State

The eutrophication status was evaluated as described in the Regulations for Lake Eutrophication Survey (2nd Edition) compiled by Jin (1990) and the Notice on the Issuance of Regulations on Evaluation Methods and Classification Techniques for Eutrophication of Lakes (Reservoirs) (CEMS, 2001) [31]. The evaluation index was the Carlson index modified by the TLI. The empirical formula of each water quality index was derived from the survey results of 26 large- and medium-sized reservoirs in China. The eutrophication process is dominantly controlled by Chla, TP, TN, SD, and CODMn. The TLI method is extensively used since it combines the simplicity of a univariate analysis with the accuracy

of a multivariate analysis [32]. This comprehensive nutrient status index (TLI) is the most suitable evaluation index of lakes in China. It is computed as follows:

$$\text{TLI} = \sum_{j=1}^{m} W_j \times \text{TLI}_j, \tag{2}$$

$$W_j = \frac{r_{ij}^2}{\sum_{j=1}^{m} r_{ij}^2} \tag{3}$$

In this formula, $W_j$ is the weight of the nutrient status index of the $j_{\text{th}}$ water quality index. $\text{TLI}_j$ is the status index of the $j_{\text{th}}$ water quality parameter, and m is the number of types of water quality indicators. $r_{ij}$ defines the correlation coefficient between Chla and indicator $j$ in Chinese lakes (reservoirs), as shown in Table 2. The correlation coefficients are tabulated in Table 1:

**Table 2.** Correlation coefficients between Chla and water quality parameters in Chinese lake (reservoirs).

| Water Quality Indicator | Chla | TP | TN | SD | COD$_{\text{Mn}}$ |
|:---:|:---:|:---:|:---:|:---:|:---:|
| $r_{ij}$ | 1.00 | 0.84 | 0.82 | −0.83 | 0.83 |

The comprehensive trophic level indices are calculated as follows:

$$\left.\begin{array}{l}
\text{TLI}_{(\text{chla})} = 10 \times (2.5 + 1.086 \ln(\text{chla})) \\
\text{TLI}_{(TP)} = 10 \times (9.436 + 1.624 \ln(TP)) \\
\text{TLI}_{(TN)} = 10 \times (5.453 + 1.694 \ln(TN)) \\
\text{TLI}_{(SD)} = 10 \times (5.118 - 1.94 \ln(SD)) \\
\text{TLI}_{(COD_{Mn})} = 10 \times (0.109 + 2.661 \ln(COD_{Mn}))
\end{array}\right\} \tag{4}$$

In each formula, the appropriate water quality index are calculated by water quality sampling concentration. The unit of TP, TN, SD, and CODMn are expressed in mg/L and Chla is given in mg/m$^3$. The nutritional status classification of lakes (reservoirs) is shown in Table 3.

**Table 3.** Classification of nutritional status of lakes (reservoirs).

| Oligotropher | Mesotropher | Eutropher | Light Eutropher | Middle Eutropher | Hyper Eutropher |
|:---:|:---:|:---:|:---:|:---:|:---:|
| TLI < 30 | 30 ≤ TLI ≤ 50 | TLI > 50 | 50 < TLI ≤ 60 | 60 < TLI ≤ 70 | TLI > 70 |

The contribution rate of water quality index $j$ to eutrophication is as follows:

$$Con_j = \frac{\text{TLI}_j \times W_j}{\text{TLI}}. \tag{5}$$

$Con_j$ is the contribution of the $j_{\text{th}}$ water quality indicator to eutrophication, given as a percentage.

### 2.3.2. Analysis Method of Land Use and Topographic Slope Structure

Referring to the research aim, the land use structure of each monitoring unit was quantified as the areal proportion of land use type $j$ in monitoring unit $i$:

$$AR\_LUC_{i,j} = \frac{S\_LUC_{i,j}}{S\_MU_i}, \tag{6}$$

where $S\_LUC_{i,j}$ is the area of land use type $j$ in monitoring unit $i$, and $S\_MU_i$ is the area of monitoring unit $i$.

The area ratio (*AR_TPI*) of the slope position index of a given topography was calculated similar to the land use structure. The calculation formula is as follows:

$$AR\_TPI_{i,k} = \frac{S\_TPI_{i,k}}{S\_MU_i},$$ (7)

where $AR\_TPI_{i,k}$ is the area proportion of a type *k* topographic position in monitoring unit *i*, and $S\_TPI_{i,k}$ is the area of a type *j* topographic position in monitoring unit *i*.

The area proportion of a type *k* topographic position with a type *j* land use type in monitoring unit *i* can be expressed as follows:

$$AR\_LUC\_TPI_{i,j,k} = \frac{S\_LUC\_TPI_{i,j,k}}{S\_MU_i},$$ (8)

where $S\_LUC\_TPI_{i,j,k}$ is the area of a type *k* topographic position with a type *j* land use in monitoring unit *i*.

In this study, the intensity of land use and development in a monitoring unit is represented by the comprehensive index of land use degree (*I*), as shown in Table 4 [33]. The higher the value of this index, the greater is the disturbance of human beings to land use. The mathematical expression is as follows:

$$I = 100 \times \sum_{i=1}^{n} A_i \times C_i,$$ (9)

where $A_i$ is the grade *i* land use degree grading index and $C_i$ is the areal percentage of grade *i* land use degree. $A_i$ divides the land use into four levels as described by Zhuang and Liu [34]. One can also refer to the research results of of Zhao et al. [35]. The assignment and calculation were performed in ArcGIS10.2.

**Table 4.** Grades of land use degree classification.

| Classification Types of Land Use | Land Use Types | Graded Index |
|---|---|---|
| Unused land grade | Unused land or unusable land | 1 |
| Natural regeneration land grade | Grassland, forest land, waters | 2 |
| Artificial regeneration of the land grade | Cultivated land, garden plot, and artificial grassland | 3 |
| Non-renewable land grade | Urban, residential areas, industrial and mining land, and land for transportation | 4 |

### 2.3.3. Analysis Methods of TP and TN Impact Factors

Since a comprehensive analysis of water quality is extremely complex, it is undertaken using a correlation analysis, which measures the degree of correlation between two variables. In this study, a Spearman correlation analysis was performed using SPSS, and the relationships between water quality and land use structure topographic position and "L-SP" were obtained via the 21 monitoring units that were obtained as samples. The explained quantity of each index to TN and TP was obtained through a redundancy analysis using Canoco 5.0. A redundancy analysis statistically evaluates the relationship between one or several variables and another group of multivariate data and can qualitatively describe the credibility and interpretation ability of the environmental change of a certain indicator. The arrows of two variables pointing in the same direction show a positive correlation and the angle between the arrows is inversely proportional to the degree of correlation between them. The lengths of arrows can be interpreted as mutual similarities of variable contributions [34].

### 2.3.4. Technical Route

The methodology of this study is shown in Figure 2 below.

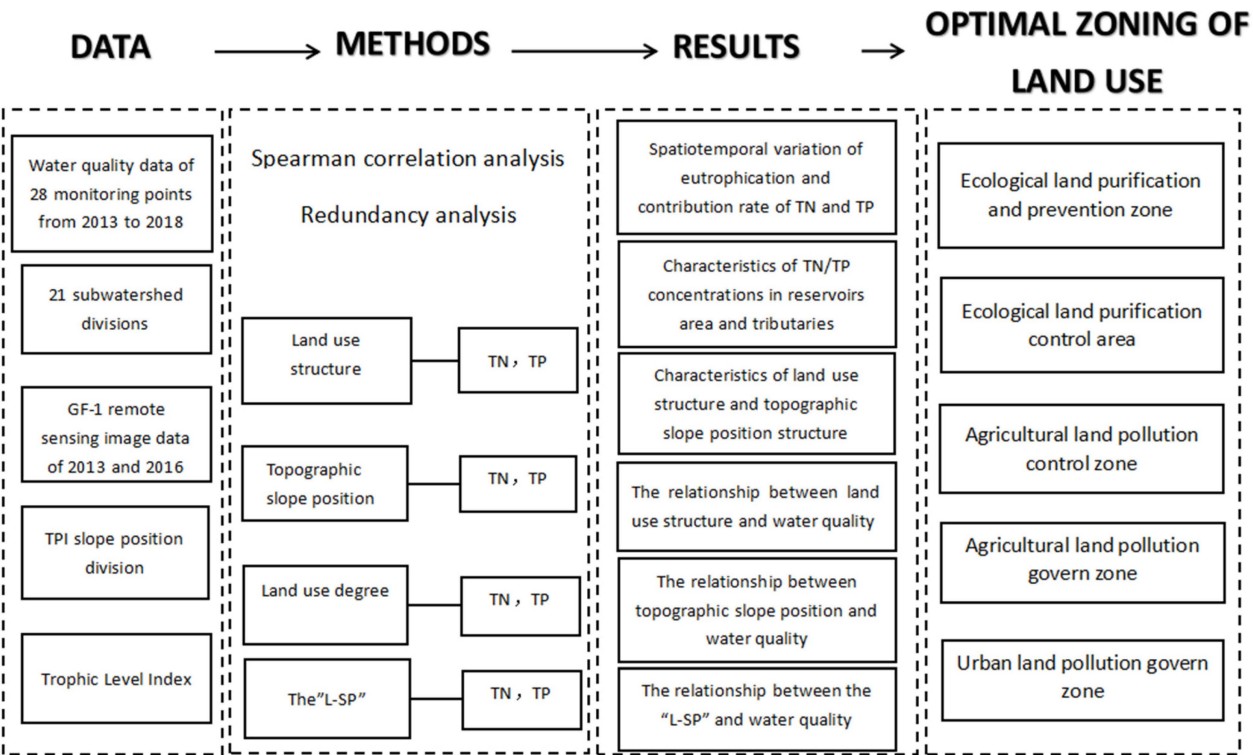

**Figure 2.** Technical route.

Based on the data from 28 water quality monitoring sites during 2013–2018 and the land use data during 2013 and 2016 in the Plateau Lakes of Guizhou Province, this paper divided the lake into 14 sub-basins and 21 monitoring units by using ArcSWAT model. The topographic position index (TPI), land use structure (LUC) and land use degree (I) were calculated by ArcGIS.

Based on the comprehensive nutrient status index (TLI), the temporal and spatial distribution characteristics of nutrient status in the HBA watershed during flood period and dry period were analyzed. The concentrations of TN and TP were compared between the reservoir area and the tributaries. In addition, the temporal and spatial variation characteristics of TN and TP were analyzed.

Spearman correlation and redundancy analyses were used to reveal the relationship among land use comprehensive topographic position and TN and TP in the lakes of Guizhou Plateau.

Finally, the principle of superimposed zoning was adopted to optimize the land use zoning, taking different management measures in different regions, providing a basis for treatment of Plateau Lake basin and other water bodies.

### 3. Results

*3.1. Change Characteristics of Eutrophication, TP, and TN Contribution Rates*

3.1.1. Time-Series Change Characteristics of Eutrophication and Contribution Rates of TP and TN

Figure 3 shows the TLI and contribution rates of TP and TN to TLI in Hongfeng Lake, Baihua Lake, and the Aha Reservoir during 2013–2018. The TLI is plotted over the whole year (January–December), during the flood season (May–October), and during the dry seasons (January–April, November–December).

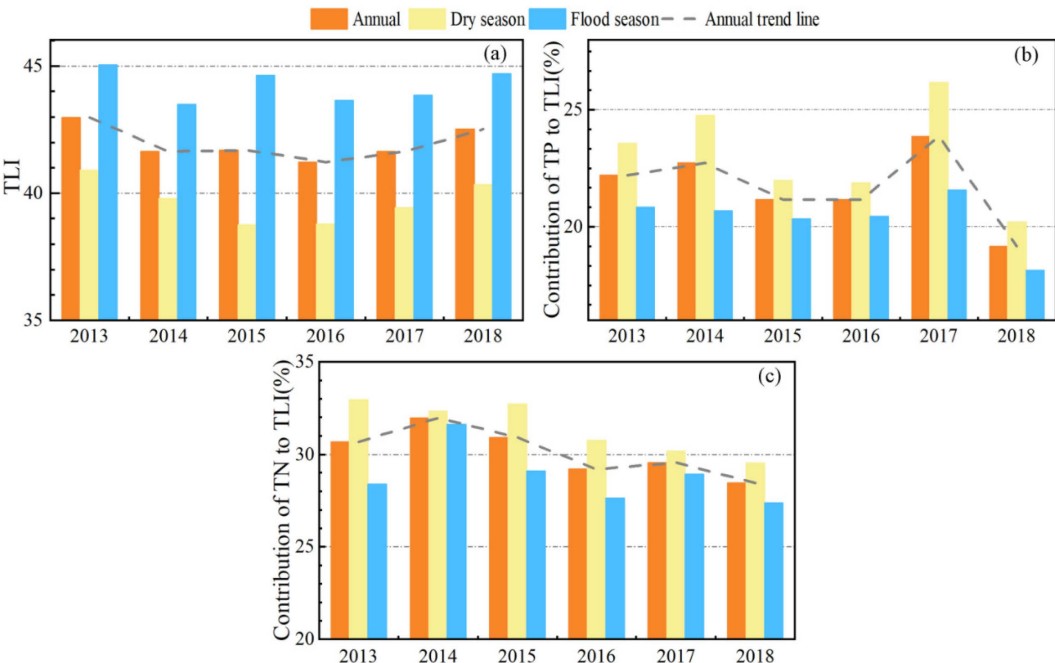

**Figure 3.** Contribution rates of annual TLI and total phosphorus (TP) and total nitrogen (TN) to trophic level index (TLI): (**a**) variation trend of annual and seasonal TLI; annual and seasonal contribution rates of (**b**) TP to TLI and (**c**) TN to TLI.

The HBA watershed was mesotrophic for six consecutive years, and the TLI value was higher in the flood season than in the dry season. The maximum and minimum TLIs were 50.70 (in the flood season) and 36.90 (in the dry season), respectively. The interannual TLI demonstrated a weakly decreasing–increasing trend. The eutrophication of water bodies in the river basin remains serious.

However, the contribution rate of TP to TLI was higher in the dry season than in the flood season. The differences in the contribution rates of TP to TLI varied between years, from 2.50% between 2015 and 2016 to 5.00% between 2015 and 2017. The annual contribution rate of TP to TLI was maximal (25.80%) in 2017 and minimal (19.20%) in 2018. Between 2013 and 2018, the ConTP demonstrated an overall fluctuation with no obvious trend. Since 2017, the contribution rate of TP has obviously declined, indicating that TP control has achieved initial results in recent years. The main method explored the eco-agricultural model for efficient utilization of soil nutrient elements using ecological principles to make farmland harvest into a variety of high-quality agricultural products. Simultaneously, green manure planting, dual use of vegetable fertilizer, and use of organic fertilizers were implemented to optimize the planting structure.

The contribution rate of TN to TLI was higher in the dry seasons than in the flood season except in 2014, when the TN contribution rates were almost equal in the wet and dry seasons. Yearly management reduced the overall contribution rate of TN to TLI from 2013 to 2018. The TN contribution rate was maximal (33.50%) in 2014 and minimal (28.50%) in 2018. The contribution rate of TN to TLI was ~5.00% higher than that of TP, and TN control has begun to play a positive role.

To quantify the eutrophication status of the major water bodies of the HBA watershed in more detail, the eutrophication and contribution rates of TP and TN to the TLI were examined on the monthly scale. The results are shown in Figure 4.

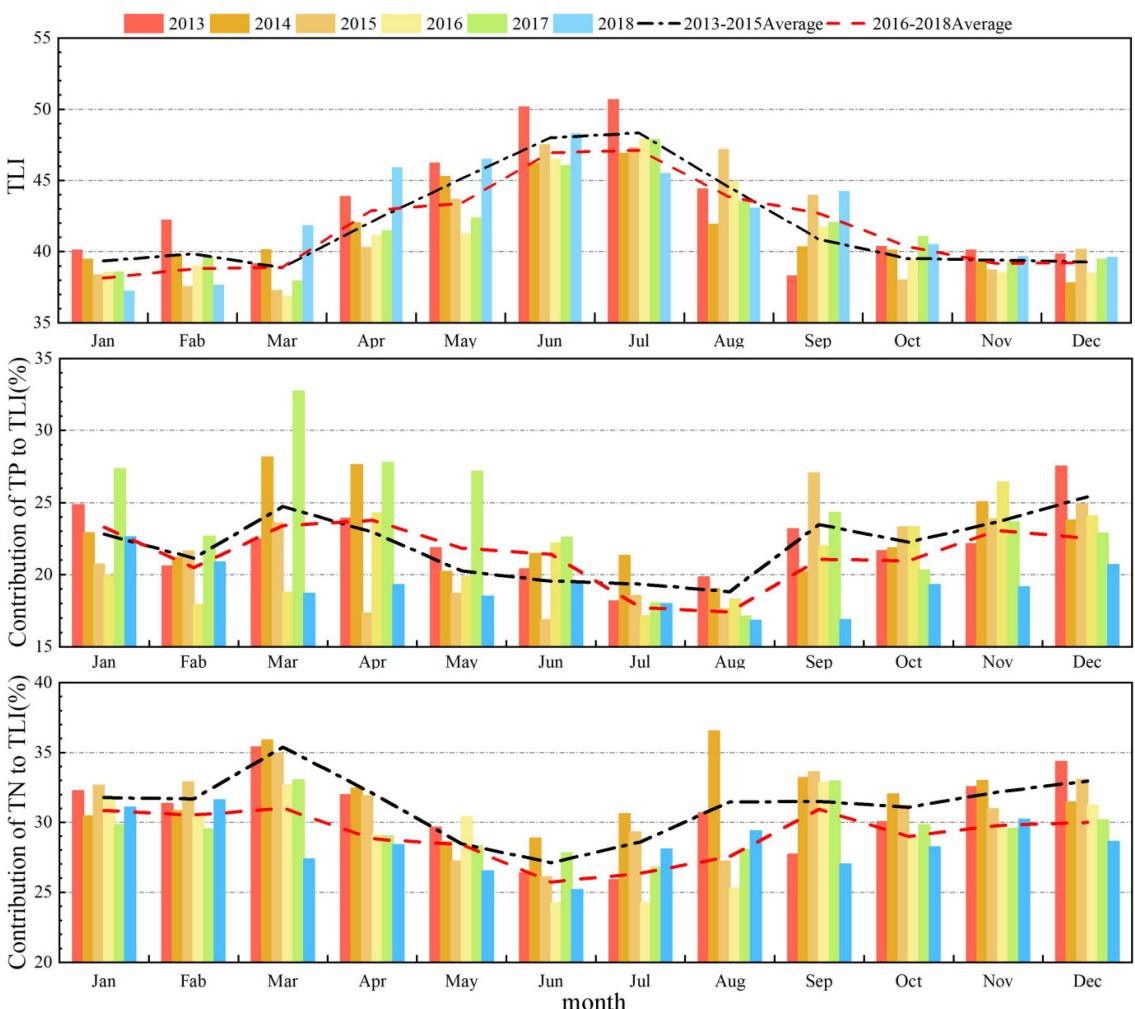

**Figure 4.** Monthly-scale TLI values (**top**) and monthly-scale contribution rates of TN (**center**) and TP (**bottom**) to TLI.

Between 2013 and 2018, the monthly TLI range ranged from 36.90 to 50.67, indicating a better nutritive state of the water body than in other areas in Guiyang [33]. The TLI was high (~50.00) from June to July 2013, indicating a higher risk of cyanobacteria outbreak. The TLI values in July largely differed between 2013 and 2018 (by ~75%), thus demonstrating a decrease followed by an increase with a minimum around 2015. The mean TLI was slightly higher in 2013–2015 than in 2016–2018 from January to August, whereas the opposite trend was observed from September to October. The TLI values in 2013–2015 were similar to those in 2016–2018 from November to December, indicating that the eutrophication risk of the HBA watershed was higher in 2013–2015 than in 2016–2018.

Between 2013 and 2018, the contribution rate of TP to TLI varied from 14.31% to 22.85%. Over this period, the contribution rate of TP to TLI from May to August was relatively low (only 17.30% at minimum). The contribution rates of TP to TLI in any given month significantly differed in different years. The maximum difference was as high as 15.50%. Moreover, the contribution rate of TP to TLI in each month was significantly lower in 2015 than in other years, and was significantly higher in January–June in 2016 than in other years. In March 2016, the contribution rate of TP to TLI reached 33.50%. During January–March and July–December, the mean contribution rate of TP to TLI was higher in 2013–2015 than in 2016–2018, indicating that the TP concentration slightly eased since treatment began in 2015.

Between 2013 and 2018, the contribution rate of TN to TLI varied from 23.33% to 31.04% on a monthly scale. In the same year, the contribution rate of TN to TLI from

May to July was generally low, with a minimum of <25.00%. Over the same month, the yearly differences in the contribution rates of TN to TLI were small and basically fluctuated at ~5.00% (reaching 12.50% at most). The average contribution rate of TN to TLI was significantly higher in 2013–2015 than in 2016–2018. During the six-year period, the contribution of TN to TLI reduced by a maximum of 5.00%, indicating that the TN concentration decreased each year under the incremental effect of TN control.

To summarize, the water in the HBA watershed demonstrated a medium trophic state from 2013 to 2018 and the annual TLI demonstrated a weakly decreasing–increasing trend. TN played an important role in the process of water nutrition in the watershed. In recent years, Guiyang has a specific administration bureau in the HBA watershed, which restricts the production, sale, and use of phosphorus-containing detergents within the river basin. This bureau comprehensively strengthens the implementation of biological purification and restoration projects. The decreasing contributions of TN and TP indicate that the treatment has achieved initial results but should be improved with additional measures.

3.1.2. Spatial Variation Characteristics of Eutrophication and Total Phosphorus and Total Nitrogen

The nutritional status of HBA watershed largely determines the drinking water safety of Guiyang citizens and is itself dominantly affected by TP and TN concentrations in the tributaries. As shown in Figure 5, from the spatial distributions of TP and TN concentrations in the reservoir area and tributaries, we can identify the influence of tributaries on the water body of the reservoir to some extent. Analyzing the different monitoring units over the 2013–2018 period, the average TLI was significantly higher in the Aha Reservoir (45.330) than in Baihua Lake (43.460) and Hongfeng Lake (40.070). The Aha Reservoir appeared eutrophic while Hongfeng Lake appeared paurophic.

The mean time of TP concentration in the monitoring points was inconsistent over the monitoring period. The TP concentrations in different seasons decreased as follows: flood season (0.040 mg/L) > annual (0.035 mg/L) > dry season (0.029 mg/L) in Hongfeng Lake, dry season (0.134 mg/L) > annual (0.124 mg/L) > flood season (0.113 mg/L) in Baihua Lake, and dry season (0.293 mg/L) > annual (0.249 mg/L) > flood season (0.205 mg/L) in the Aha Reservoir. The Aha Reservoir and Baihua Lake are rainfall-replenished and their water storage is depleted as rainfall runoff decreases during the dry seasons. TP is a sedimentary nutrient, and so is more concentrated in the Aha Reservoir and Baihua Lake during the dry season than in the flood season. The annual, dry season, and flood season spatial distributions of TP concentration had minor variations in each monitoring unit. For example, in the flood season, the spatial distributions of TP concentration were 0.224–0.599 mg/L in the Aha Reservoir, 0.027–0.396 mg/L in Baihua Lake, and 0.026–0.077 mg/L in Hongfeng Lake. The TP concentration was significantly higher in the Aha Reservoir and the Baihua Lake tributaries than in the reservoir area. The ratio of TP concentrations between the tributaries and reservoir area reached 14.44 at maximum. The high TP concentration carried by the tributaries was an important cause of increased TP concentration in the reservoir area.

During each monitoring period, the overall mean TN concentration decreased in the order of flood season > annual > dry season. The TN concentration of the HBA watershed was generally high ranging from 1.200 to 10.750 mg/L during the flood season and from 0.920 to 18.700 mg/L during the dry season. The maximum TN concentration was 18 times higher than the Class III TN concentration (1.000 mg/L) in the Environmental Quality Standard for Surface Water (GB3838-2002). The spatial distributions of TN concentration during the two seasons were almost the same in different monitoring units. For example, during the flood season, the TN concentration was highest in the Aha Reservoir (5.106 mg/L), intermediate in Baihua Lake (3.759 mg/L), and lowest in Hongfeng Lake (1.870 mg/L). Comparing the spatial and temporal distributions of TP in Hongfeng Lake, the TN concentration was higher in the tributaries of Hongfeng Lake than in the reservoir area.

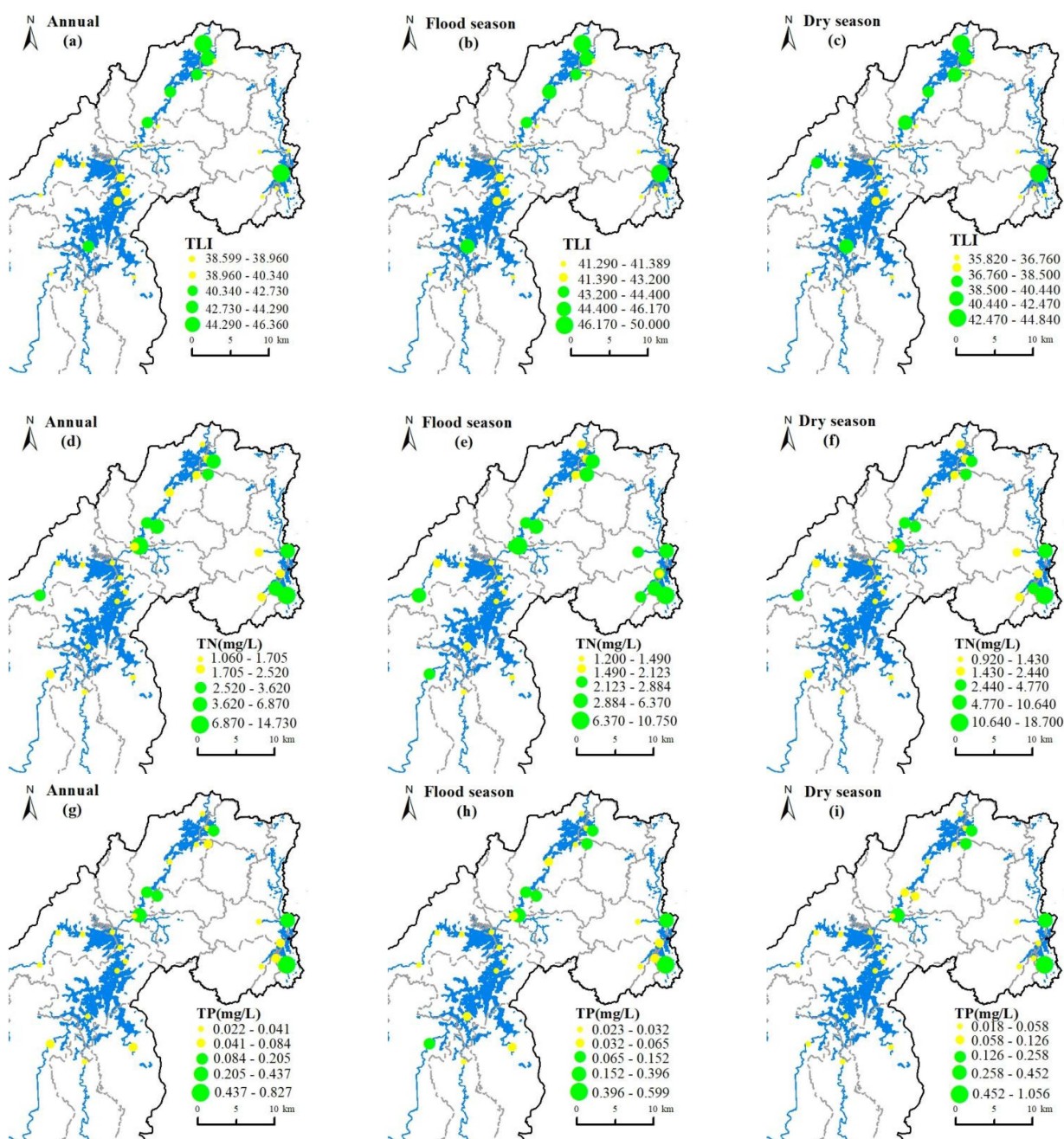

**Figure 5.** Spatial distributions of TLIs (**top row**), TN concentrations (**center row**), and TP concentrations (**bottom row**). (**a**,**d**,**g**) Annual; (**b**,**e**,**h**) flood Season; (**c**,**f**,**i**) dry season.

Between 2013 and 2018, the nutrient concentration in the HBA watershed demonstrated obvious spatial differences and the TP concentration increased with the river trend (i.e., the TP concentration was significantly higher in the downstream Aha Reservoir than in the upstream Hongfeng Lake Basin). Considering the differences in land use among different river basins, Hongfeng Lake that is located in the upper reaches of the river is used for various purposes, such as drinking water source, power generation, and tourism. In addition, the land types on both sides of Hongfeng Lake basin are paddy fields. Rice is the main crop and sporadic coal mines are found near the upper reaches. Baihua Lake is located downstream of Hongfeng Lake; thus, it is more polluted than Hongfeng Lake. However, water quality is of great concern in the Aha Reservoir since sewage and garbage are discharged into the Aha reservoir and the peasants hold agritainment, a kind of farm-based

tourism. Note that the TN concentration is considerably affected by land use, particularly by agricultural activities. Different terrestrial nutrients are carried by surface and underground runoff, leading to different spatial and temporal distributions of nitrogen nutrients. Considering the spatial differences of water nutrients in river basins, the flow characteristics of nutrients in different regional spaces must be systematically identified and the land use control areas must be rationally divided.

### 3.1.3. Effect of Nitrogen-to-Phosphorus Ratio on Eutrophication in Water Bodies

The TN/TP ratio shows the effect of nutrient input on the nutrient structure of a water body. If the TN/TP ratio exceeds 16:1, the primary influencing factor is usually phosphorous [32]. In the HBA watershed, the TN/TP ratio was between 28.2 and 153.4 (Figure 6), indicating that phosphorus is the primary influencing factor and present at low levels in the watershed. Therefore, controlling phosphorous content is the key to inhibiting eutrophication of the HBA watershed. Moreover, the TN/TP ratios at different monitoring sites demonstrated large seasonal variations and a decreasing trend from upstream to downstream. Since Hongfeng Lake is located at the upstream of the watershed, it directly receives water from the tributaries, and so its TN/TP was highest among the sites. However, the Aha Reservoir presented the lowest N/P ratio among the sites. During the dry seasons, the TN/TP in HongFeng Lake was maximized at the HouWu site and minimized at the Sancha River site. In Baihua Lake, the maximum and minimum TN/TP ratios were reported at the Maxi River and Huaqiao sites, respectively. Furthermore, in the Aha Reservoir area, they appeared in the middle and east of the reservoir area, respectively. During the flood season, the TN/TP was maximum at the PianShanZhai site of Hongfeng Lake and at the BaiDaba and MaXi River sites of Baihua Lake. In the Aha Reservoir, the TN/TP was maximized in the middle of the reservoir area as observed in the dry seasons. Along with the TN and TP concentrations at each monitoring point, these data show that the Sancha River and HuaYu in Hongfeng Lake and Huaqiao in the Baihua Lake area were considerably polluted by phosphorus, facilitating the accumulation of nutrients in the HBA watershed. The high TN/TP ratio in Hongfeng Lake during the dry seasons indicates a lower eutrophication probability in the HBA watershed during these periods than during the flood season. The Baihua Lake and Aha Reservoir exhibit the opposite trends, indicating that they are probably eutrophic during the flood season.

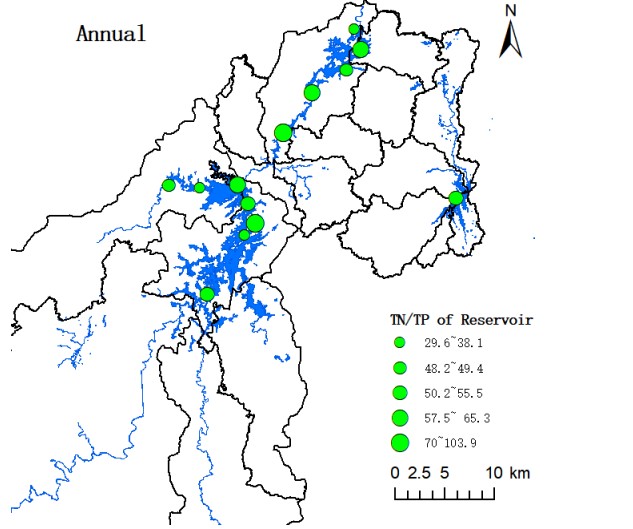
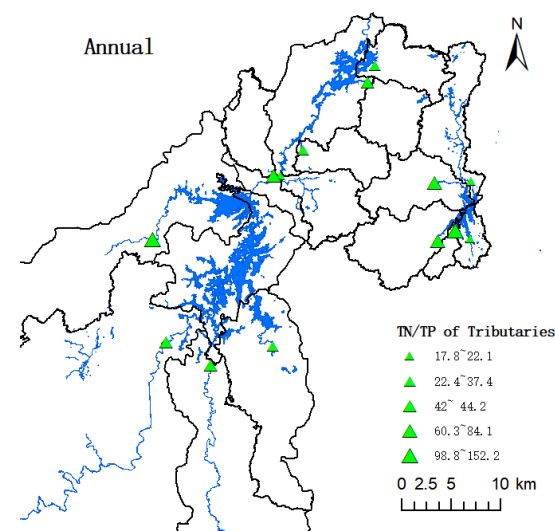

**Figure 6.** *Cont.*

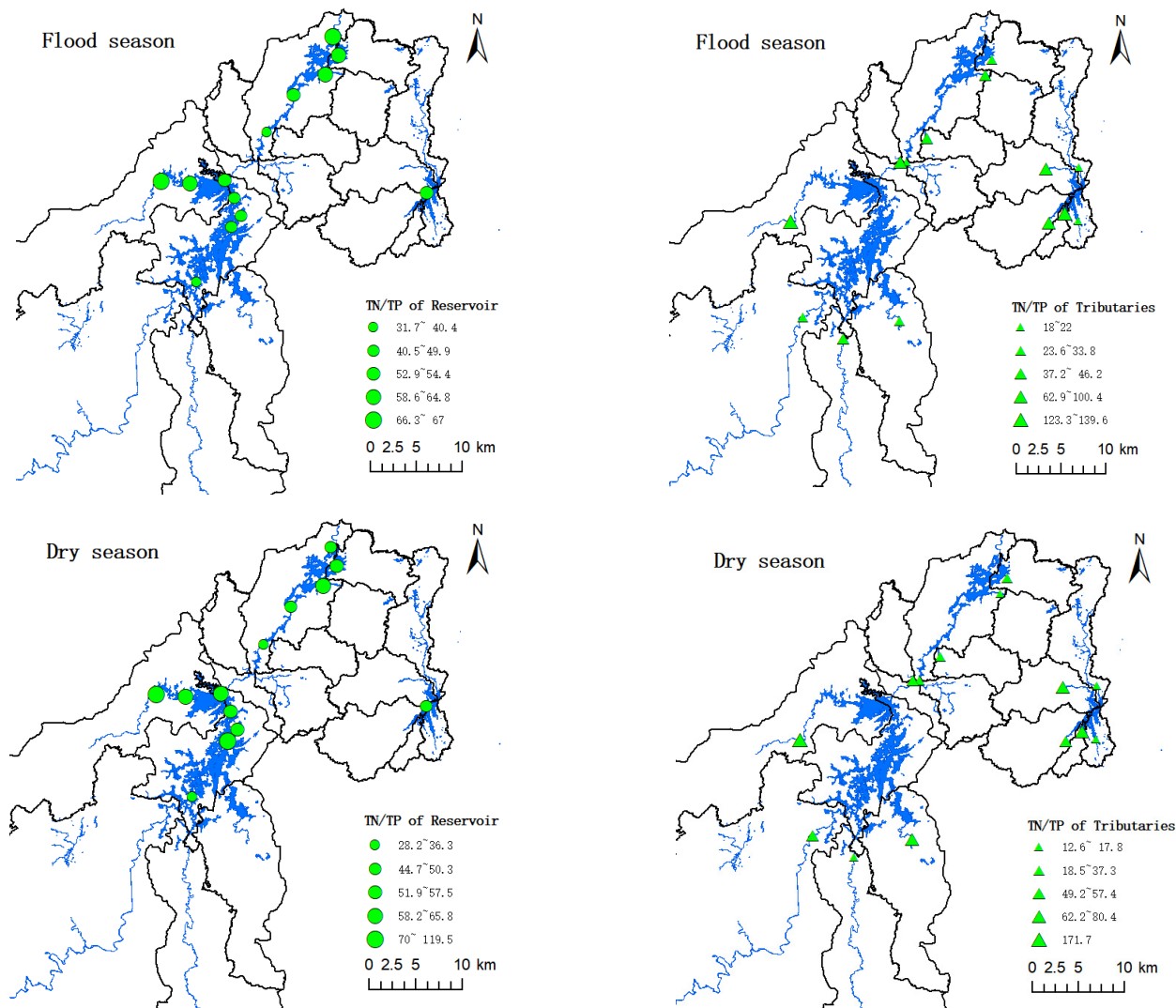

**Figure 6.** Nitrogen-to-phosphorus ratios at different monitoring points during different seasons in the reservoir area (**left**) and tributaries (**right**).

Therefore, relevant departments should focus on water quality in the SanCha River of Hongfeng Lake, Huaqiao of Baihua Lake, and the eastern Aha Reservoir. Controlling the amount of phosphorus upstream of the reservoir is important for preventing additional spread of the pollution. The N/P ratio is higher in other parts of Hongfeng Lake than in the Baihua Lake and Aha Reservoir, indicating a relatively high water quality and certain achievements of recent environmental protection. Follow-up protection work should be managed in a timely manner to ensure the water safety of Guiyang's urban residents.

3.1.4. Comparison of TN and TP Concentrations in the Tributaries and Reservoirs

Comparison of TN and TP Concentrations in the Tributaries and Reservoirs was shown in Figure 7.

The TN concentration at each monitoring point was higher in the tributaries than in the reservoir. The highest and lowest TN levels were reported at the Huaqiao site of Baihua Lake and the XiJiao reservoir of Hongfeng Lake, respectively. The average TN levels in the three reservoir areas decreased in the following order: Aha Reservoir (2.172 mg/L) > Baihua Lake reservoir (2.075 mg/L) > Hongfeng Lake reservoir (1.362 mg/L). Hongfeng Lake is moderately eutrophic, whereas Baihua Lake and Aha Reservoir are slightly eutrophic. In the tributaries, the average TN concentration decreased in the order of Baihua Lake

(6.15 mg/L) > Aha Reservoir (4.92 mg/L) > Hongfeng Lake (2.11 mg/L). The highest and lowest levels appeared at the LanNiGou site of Aha Reservoir and the MaXian River site of Hongfeng Lake, respectively.

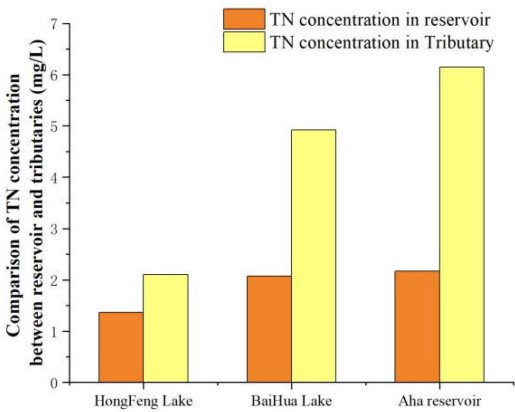
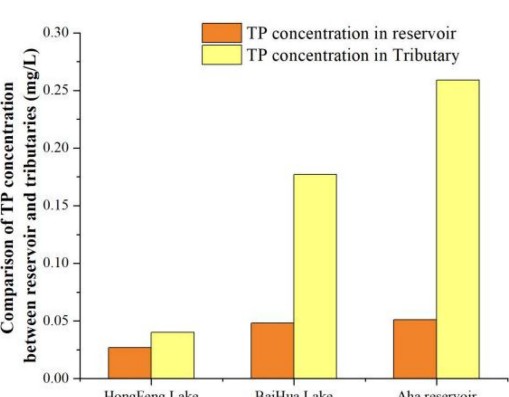

**Figure 7.** TN (**left**) and TP (**right**) concentrations in the reservoir and tributaries.

The TP concentration was lower than the TN concentration but was higher in the tributaries than in the reservoir areas. The TP concentration was highest in the east of the Aha Reservoir area and lowest at the DaBa and PianShanZhai sites of Hongfeng Lake. The average annual average TP concentrations in the three reservoirs follow the same trend as the TN concentration, being highest in the Aha Reservoir (0.051 mg/L), intermediate in Baihua Lake (0.048 mg/L), and lowest in the Hongfeng Lake reservoir (0.027 mg/L). In the tributaries, the highest and lowest TP concentrations appeared at the LanNiGou site of the Aha Reservoir and the Maiweng River site of Hongfeng Lake, respectively. The TP concentration decreased in the following order: Aha Reservoir tributary (0.259 mg/L) > Baihua Lake tributary (0.177 mg/L) > Hongfeng Lake tributary (0.05 mg/L). During the study period, the TN and TP concentrations in the tributaries far exceeded the internationally recognized important concentration of eutrophication [32] and were suitable for algal growth. These high TN and TP concentrations improved the eutrophication state of the HBA watershed and degraded the overall water quality.

In addition to the TN and TP concentrations, the eutrophication states should be compared in the reservoir and tributaries. The calculated TLI in the tributaries of Hongfeng Lake was 52, indicating a mildly eutrophic water quality. The TLIs in the tributaries of the Baihua Lake and Aha Reservoir were 67 and 68, respectively, and therefore the water quality in both tributaries is severely eutrophic. The TLIs were lower in Hongfeng Lake (47), Baihua Lake (52), and the Aha Reservoir (57). Therefore, regardless of TN versus TP level, the water quality at each monitoring point was lower in the tributaries than in the reservoir. As the tributaries flow through multiple areas, they are continually polluted by nutrients accumulated from urban and rural living and industrial waste water.

3.1.5. Comparison of TN and TP Contribution Rates to Eutrophication in Tributaries and Reservoirs

As shown in Figures 8 and 9, based on the contribution rate of TN to TLI, the ConTN value in the Aha Reservoir is 4.23% higher in the inflow branch than in the reservoir. This result is related to the high annual average concentration of TN (6.146 mg/L) in the tributaries of the Aha Reservoir. However, Hongfeng Lake is located in the upper reaches of the river, where pollution is relatively low. There are some agricultural activities along the of Hongfeng Lake, where rice is the main crop and sporadic coal mines are found near the upper reaches, which lead to the generation of most of the living sewage along the lake. In addition, industrial waste water around Baihua Lake and agricultural practices directly discharge waste water into the reservoir area; therefore, the ConTN values are higher in the reservoirs of Baihua Lake and Hongfeng Lake than in the tributaries. The relocation of key

polluters in the lake region can be intensified. In addition, and the discharge of wastewater from hydropower plants can be limited in order to purify the water body. Moreover, the ConTN values of both the Aha tributaries and reservoir are higher during the flood season than during the dry season. During the flood season, multiple pollutants such as nitrogen- and phosphorus-rich fertilizer are carried with the increased river flow, thus increasing the water eutrophication.

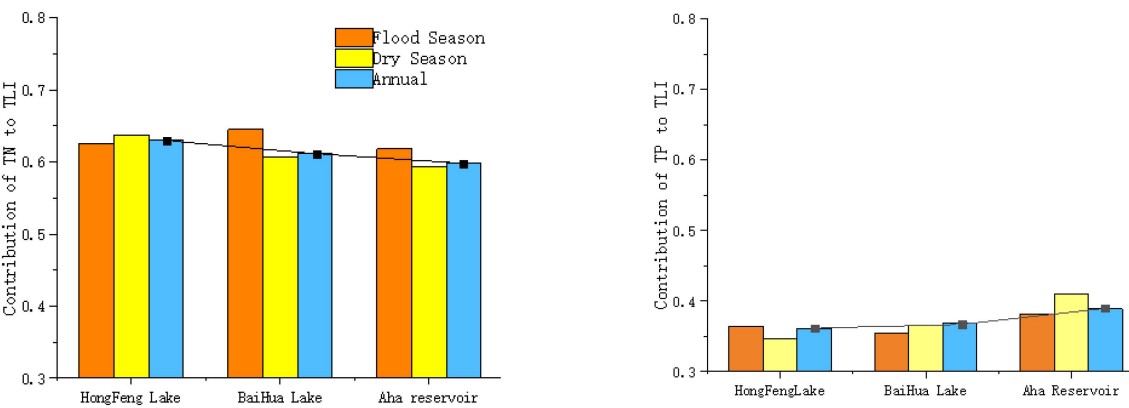

**Figure 8.** Contribution rate of TN (**left**) and TP (**right**) to TLI in the reservoirs.

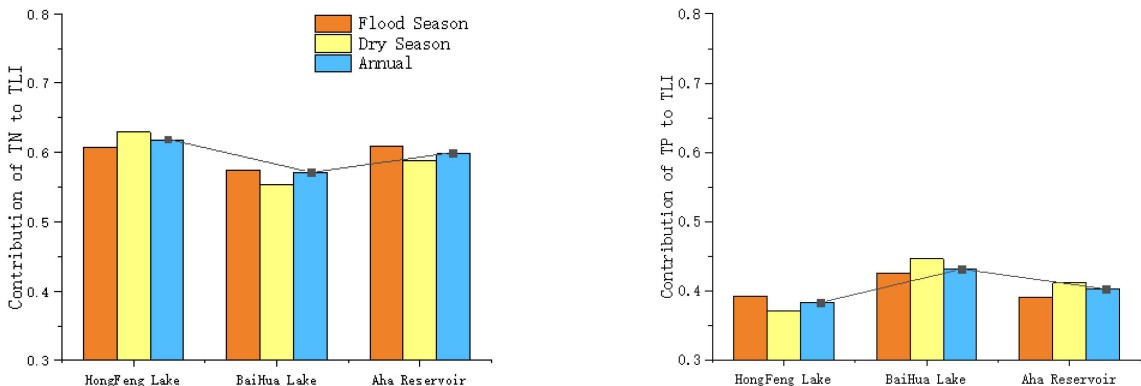

**Figure 9.** Contribution rates of TN (**left**) and TP (**right**) to TLI in the tributaries.

The TP contribution rate to the TLI is higher in the tributaries than in the reservoir. The ConTP is ~2% higher in Hongfeng Lake and Baihua Lake than in their corresponding reservoirs, and ~7% higher in the Aha Reservoir than in its surrounding area. Moreover, the ConTP values in the Baihua Lake and Aha Reservoir areas are higher in the dry seasons than in the flood season. However, the ConTP is obviously lower than the ConTN at all monitoring sites. The TP is primarily sourced from agricultural and industrial waste water.

In follow-up work, relevant departments should strengthen water quality monitoring of the LiJiaChong River and YangChang River estuaries to ameliorate agricultural non-point source pollution. This can be achieved most effectively by reducing the amount of chemical fertilizer, strategically applying fertilizer (e.g., organic fertilizers instead of synthetic and highly polluting fertilizers), reducing the arable land unsuitable for cultivation (which refer to the cultivation method with large fertilizer application), and improving the utilization rate of chemical fertilizer such as promoting the cultivation of legumes. The development of ecological agriculture would reduce the nutrient input, particularly by reducing the amount of nitrogen and phosphorus fertilizers.

### 3.2. Land Use Structure and Topographic Slope Structure Characteristics

3.2.1. Characteristics of Land Use Structure

In the early period of economic development, land use was extensive and greatly affected by human activities. With the influence of the policies of returning farmland to forests and reclaiming land, ecological projects, such as afforestation were implemented. Influenced by a series of policies, various changes in land-use have occurred.

Figure 10 compares the spatial distributions of land uses and the area proportions of different land use types in each monitoring unit of the HBA watershed in 2013 and 2016. Grassland and urban land demonstrated the greatest changes between 2016 and 2013, increasing by 8.42% and decreasing by 0.11%, respectively. Among the monitoring units, the ecological (agricultural) land increased (decreased) by 9.16% (3.30%) at Hongfeng Lake, by 5.59% (11.32%) in Baihua Lake, and by 11.61% (7.89%) in the Aha Reservoir, where the urban land decreased by 2.98%. Moreover, the land use degree from 2013 to 2016 was highest in the Aha Reservoir, intermediate in Baihua Lake, and lowest in Hongfeng Lake, with average comprehensive indexes of land use degree of 263.18, 252.74, and 247.74, respectively. The total proportion of land use change was 58.61% in the HBA watershed, 58.42% in Hongfeng Lake, 59.32% in Baihua Lake, and 58.93% in the Aha Reservoir. Although the areas of urban and agricultural lands are currently declining, development activities in both land types have remained strong over the historical period via the rapid transfer of land use. The land use changes are drastic in both the HBA watershed and its sub-basins (Hongfeng Lake, Baihua Lake, and the Aha Reservoir).

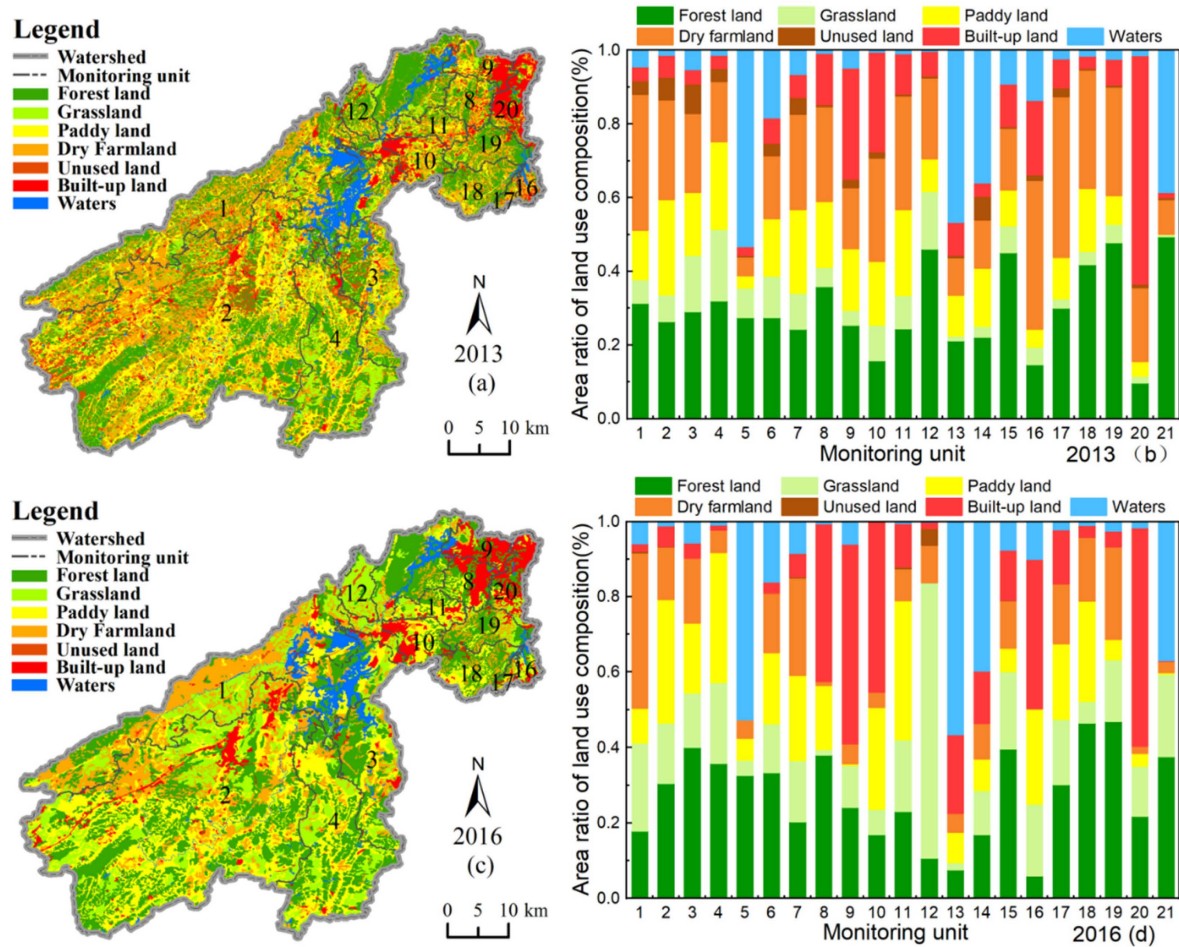

**Figure 10.** Spatial distributions of land use types and land use composition of the monitoring units in 2013 (**a,b**) and 2016 (**c,d**).

The change of land use structure in the HBA watershed is closely related to water pollution prevention and the control measures taken by the local government. In the first- and second-level protection areas of the Hongfeng Lake drinking-water source area, rural settlements were relocated to reduce the volume of domestic sewage entering the lake, the agricultural planting structure was adjusted to reduce the agricultural land coverage, the use of agricultural fertilizer was restricted to lower the agricultural non-point source pollution, and the ecological-land coverage was increased. The local government has focused on greening the construction, building livable environments, and constructing and protecting city parks.

### 3.2.2. Topographic Slope Structure Features

Figure 11 shows the proportions of slope areas in the HBA watershed. The topographical proportions are 50.14% for gentle slopes, 37.56% for steep slopes, 9.28% for valleys, and 3.03% for ridges. The whole basin is dominated by gentle slopes. From the perspective of monitoring units, the area proportions of gentle and steep slopes are 55.50% and 36.43%, respectively, at Hongfeng Lake, 39.33% and 31.57%, respectively, at Baihua Lake, and 44.41% and 35.10%, respectively, at the Aha Reservoir. Baihua Lake and the Aha Reservoir include valley slopes with area proportions of 21.58% and 19.51%, respectively. The valleys are primarily distributed in the middle part of Baihua Lake, the southeast low-lying part of the Aha Reservoir, and the northern part of Hongfeng Lake Basin, which is the tributary of Baihua Lake and the Hongfeng Lake of the Aha Reservoir. The steep slopes are primarily distributed in Baihua Lake and the Aha Reservoir with relatively large slopes, and in the northwest and southeast of Hongfeng Lake. Ridges are distributed over small areas and are concentrated in high-altitude areas.

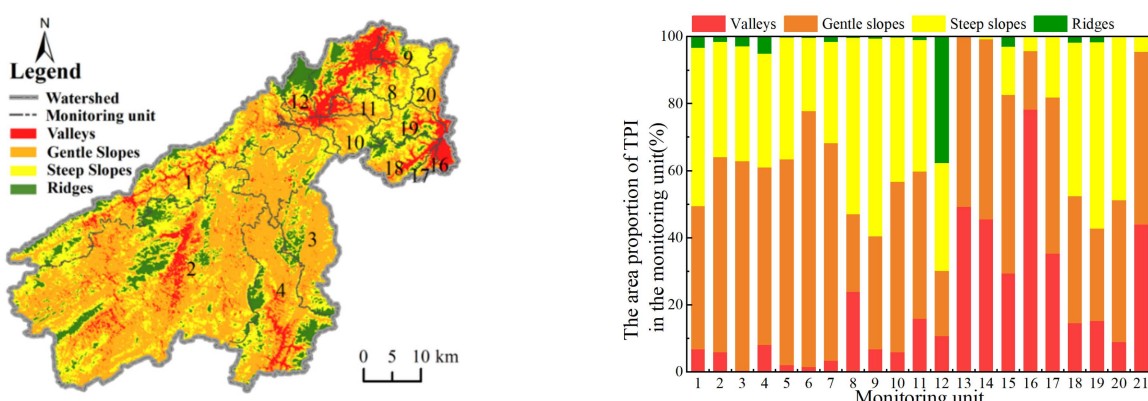

**Figure 11.** Spatial distribution of topographic position and topographic position composition in each monitoring unit.

### 3.2.3. Features of Slope Position Compound Land Use Structure

Since unused land is not distributed in most of the monitoring units, the complex topographic slope areas in this study were divided into five land types: woodland, grassland, paddy land, dry farmland, and construction land, which are plotted as five bar charts for each monitoring site in Figure 12. The proportions of steep-slope forest land, gentle-slope paddy land, and gentle-slope forest land are 16.00%, 16.89%, and 11.56%, respectively. The proportions of ridge construction land, ridge dry farmland, and ridge paddy land are small (0.01%, 0.20%, and 0.04%, respectively). The areas of different topographies and slope positions of compound land use in the three sub-watersheds (Hongfeng Lake, Baihua Lake and the Aha Reservoir) are similar to those of the HBA watershed. In the early period of economic development, land use was extensive and greatly affected by human activities. Then, under the influence of the policy of returning farmland to forests and reclaiming land, ecological projects such as afforestation were implemented. Considering the changes in land use compound topographic position between 2013 and 2016, the dry farmland on

gentle slopes with a large area change in the HBA watershed decreased by 4.90% and the grassland on steep slopes increased by 4.71%. In Hongfeng Lake, the grassland on steep slopes increased by 5.29% and the dry farmland on gentle slopes decreased by 4.75%. In Baihua Lake, the dry farmland on steep and gentle slopes decreased by 6.34% and 5.00%, respectively. At the Aha Reservoir, the dry farmland on gentle slopes decreased by 6.10% and grassland on gentle slopes increased by 3.43%.

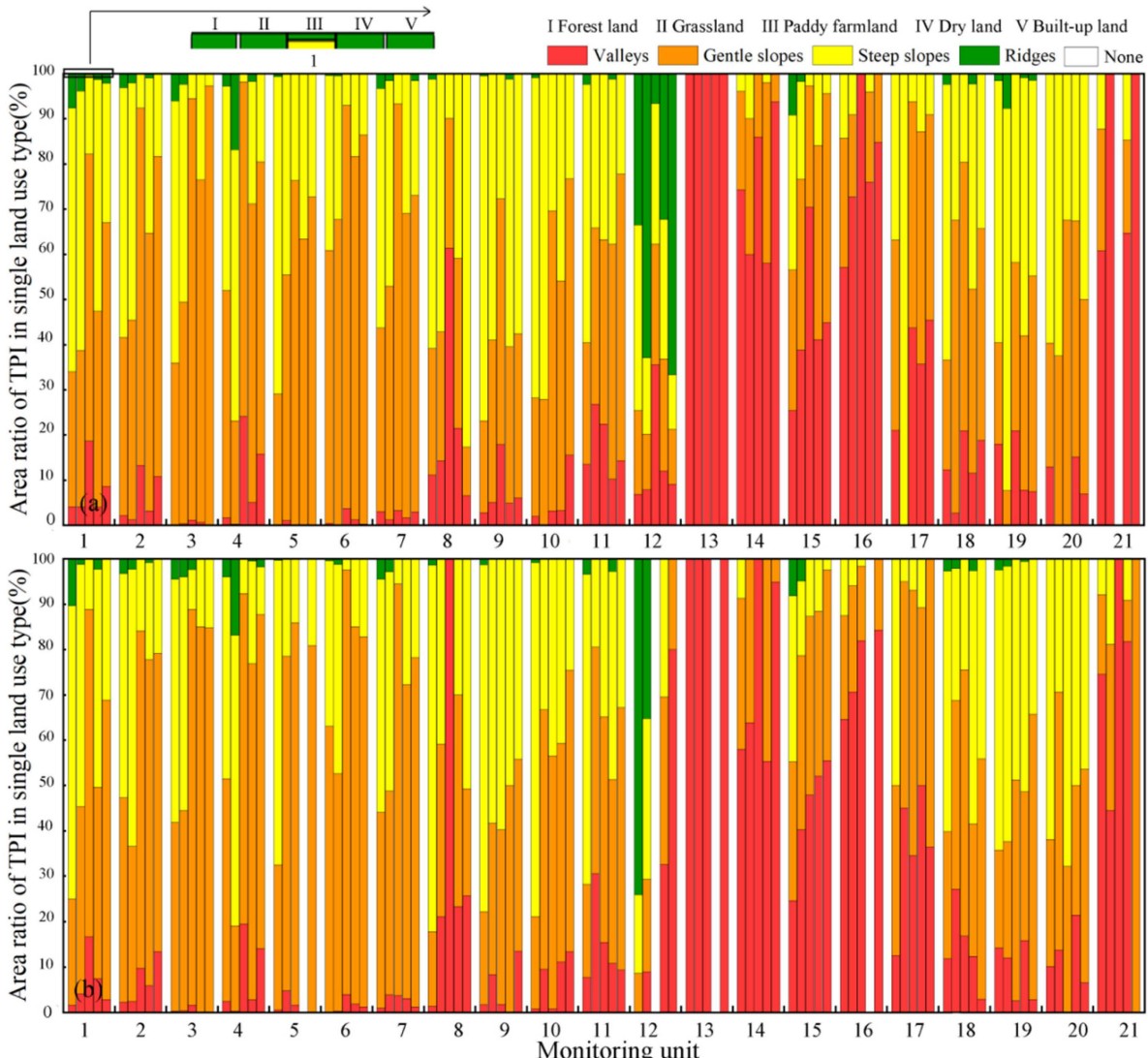

**Figure 12.** TPI compositions of land use types in the monitoring units: (**a**) 2013 and (**b**) 2015.

By superimposing the scales of topographic position and land use type in space, we can accurately determine the land use structure of the lake basin in the mountainous areas of the plateau and provide detailed information on the land structure characteristics. This analysis can facilitate regional prevention and control of water nutrients in the river basin and the determination of nutrient sources. Moreover, it provides a reference for identifying important control areas in the river basin.

### 3.3. Relationship between Land Use Types and Water Quality

3.3.1. Correlation between Land Use Types and Water Quality

The area proportions of single land use type, land use degree, and TP and TN concentrations in the 21 monitoring units of HBA watershed were correlated in two periods: 2013–2015 and 2016–2018. Figure 13 shows the Spearman correlations between the proportion of land use area of the HBA Watershed and TP and TN (where $p < 0.05$ denotes high

significance and $p < 0.1$ denotes significance). As the unused land proportion was <1.00% in >50% of the monitoring units, the correlations between TP and TN and unused land were not representative and will not be discussed.

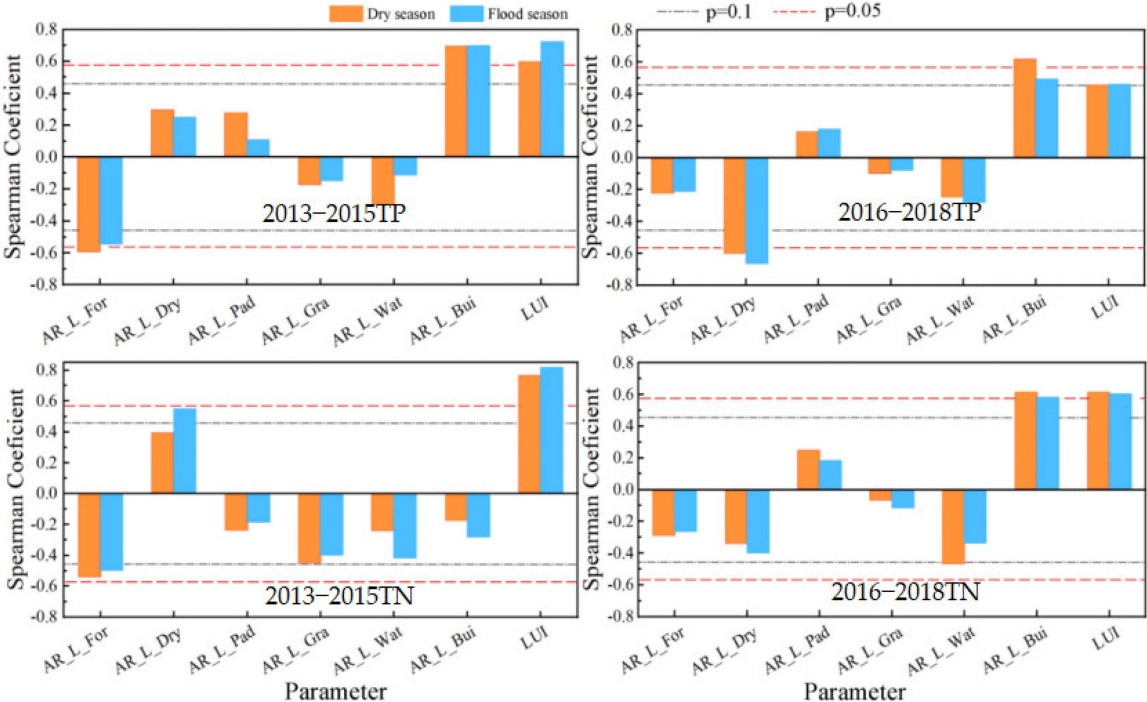

**Figure 13.** Correlations between land use structure and TP and TN concentrations.

Significant correlations between forest land and TP ($-0.543$ in the flood season; $-0.591$ in the dry seasons) and between forest land and TN ($-0.495$ in the flood season; $-0.537$ in the dry season) appeared from 2013 to 2015, but these obvious correlations disappeared from 2016 to 2018. Allocating forest land as ecological land exerted a prominent effect when the TP and TN concentrations were high.

From 2013 to 2015, dry farmland was positively correlated with TN in the flood season (0.548) and with TP in the flood and dry seasons (0.249 and 0.295, respectively). From 2016 to 2018, dry farmland was negatively correlated with TP in the flood and dry seasons ($-0.662$ and $-0.598$, respectively) and with TN in the flood and dry seasons ($-0.395$ and $-0.338$, respectively). Fertilizer application in agricultural land was the driving factor of the increased TP and TN concentrations. However, the control of chemical fertilizers and pesticides in the study area has been strengthened in recent years, (e.g., restricting recreational activities along lakes (reservoirs), prohibiting the use of chemical fertilizers in large agricultural activities, adjusting the structure of agricultural production, and developing modern and organic agriculture; thus, weakening the relationships among dry farmland, TP, and TN concentrations.

From 2013 to 2015, construction land was significantly positively correlated with TP (0.698 in the flood season; 0.695 in the dry season) and with TN (0.581 in the flood season; 0.617 in the dry season). From 2016 to 2018, the correlation between construction land and TP remained significant (0.490 in the flood season; 0.617 in the dry season) but that between construction land and TN failed the significance test by a small margin (0.458 in the flood season; 0.399 in the dry season). Construction land exerted a stable positive effect on TP and TN. In other words, the TP and TN concentrations increased with increasing construction land area.

From 2013 to 2015, land use degree was significantly positively correlated with TP in both the flood and dry seasons (0.723 and 0.595, respectively) and with TN in the flood and dry seasons (0.818 and 0.764, respectively). These correlations remained significant in 2016

to 2018: land use degree versus TP = 0.460 in the flood season, 0.454 in the dry season; land use degree versus TN = 0.604 in the flood season, 0.612 in the dry season. High-intensity urbanization and agricultural development activities can explain an increase in TP and TN concentrations.

Paddy land, grassland, and water were not significantly correlated with TP or TN. Note that grassland and water bodies were negatively correlated with TP and TN during both flood and dry seasons in both periods (i.e., grassland and water bodies tend to purify ecological land and reduce the TP and TN concentrations). The results of woodland, dry farmland, and construction land are consistent with those of Ju et al. [36].

### 3.3.2. Redundancy Analysis of Land Use Type and Water Quality

Redundant analysis ranking can accurately describe the relationship between land use types and TP and TN in the flood and dry seasons (Figure 14, Table 5). The total interpretation rate of the area ratio of land use type in HBA watershed to TP and TN was 61.4% in 2013–2015 but reduced to 25.4% in 2016–2018, indicating that the land use structure exerted a stronger influence on TP and TN in 2013–2015 than in 2016–2018. From 2013 to 2015, forest and construction land explained 38.30% and 35.40% of the water quality, respectively, and 23.50% was contributed by dry farmland. From 2016 to 2018, construction land accounted for 65.80% of the water quality, followed by dry farmland with 22.10% and woodland with 16.30%.

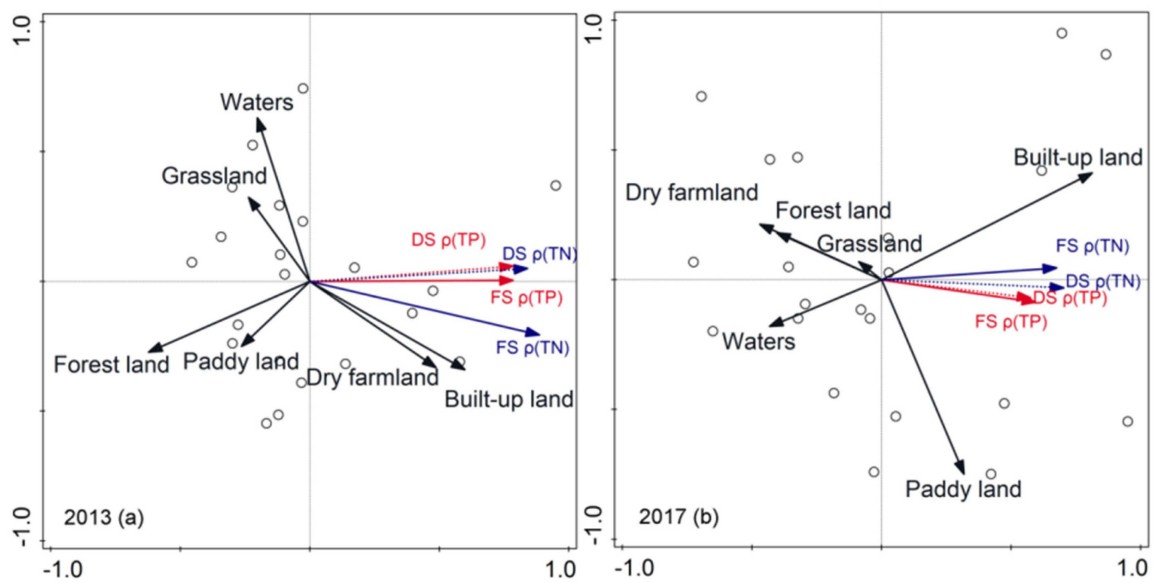

**Figure 14. Redundancy analysis of land use types and TP and TN**: (**a**) 2013 and (**b**) 2017. Solid (dotted) black, blue, and red lines represent land use types, TP, and TN, respectively, in the flood (dry) season.

**Table 5.** Explanation of TP and TN by land use types.

|  | Forest Land | Dry Farmland | Paddy Land | Grassland | Water | Built-Up Land |
|---|---|---|---|---|---|---|
| 2013–2015 | 38.3 | 23.5 | 7.0 | 5.7 | 4.6 | 35.4 |
| 2016–2018 | 16.3 | 22.1 | 10.3 | 0.8 | 18.7 | 65.8 |

To summarize, different land use types on the river basin exerted different influences on the TN and TP distributions. The maximum correlations between the area proportions of different land use types and the TN and TP concentrations in water bodies decreased in the following order: construction land (0.698) > upland land (−0.662) > woodland (−0.591) > grassland (−0.448). Forest land demonstrated a significant purification effect

on TN and TP in the water bodies, whereas construction land was a pollution source. The pollution from land banks mostly entered the tributaries and reservoir areas through precipitation and runoff. Therefore, the influences of these land use types should be considered when classifying watershed control areas.

### 3.4. Relationship between Topographic Position and Water Quality

3.4.1. Correlation Analysis between Topographic Position and Water Quality

Figure 15 compares the correlations between the topographic positions of the HBA Watershed and TP and TN levels during the 2013–2015 and 2016–2018 periods. From 2013 to 2015, the correlation between valley and TP was significant (0.538) during the dry season. From 2016 to 2018, the valley was positively correlated with TP in both the flood and dry seasons (0.562 and 0.686, respectively) and with TN in both the flood and dry seasons (0.600 and 0.645, respectively). In 2013–2015, significant correlations were reported between gentle slope and TP (−0.538 and −0.648 in the flood and dry seasons, respectively). The correlations between gentle slope and TN in the flood and dry seasons (−0.455 and −0.473, respectively) only passed the significance test. Among the types of topographic slope, valley exerted the strongest positive effect on TP and TN concentrations, indicating that TP and TN aggregate in valleys. Gentle slopes do not obviously affect the TP and TN concentrations and demonstrate no obvious source and sink characteristics. Although steep slopes and ridges were not clearly related to TP and TN in this study, most studies on agricultural non-point source pollution regard steep slopes as pollution sources. In control measures of lake eutrophication, particularly in Karst plateau mountainous areas, the effects of topographic position on water quality indices cannot be ignored.

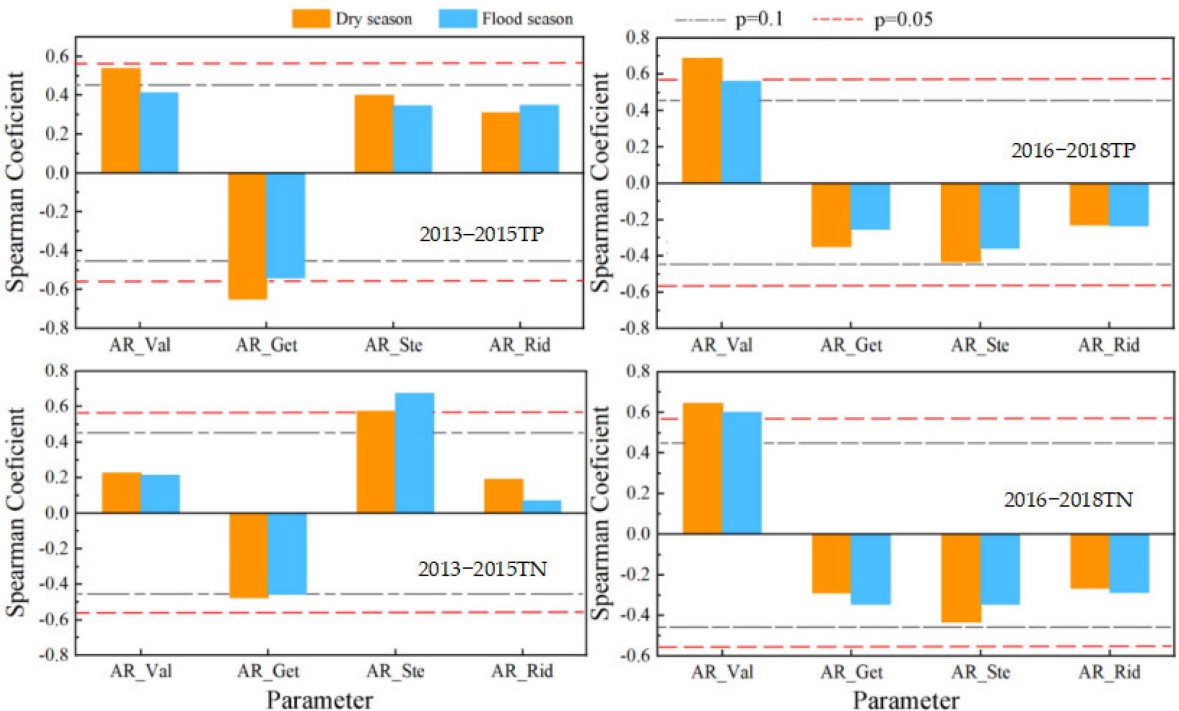

**Figure 15.** Correlations between topographic positions and TP and TN concentrations.

3.4.2. Redundancy Analysis of Topographic Position and Water Quality

The interpretation rates of topographic slope level to the TP and TN concentrations during 2013–2015 and 2016–2018 are presented in Figure 16 and Table 6. In 2013–2015, valleys in the HBA watershed were obviously related to TP and TN whereas gentle and steep slopes exerted a low influence. The cumulative explanatory amount of topographic slope level to TP and TN in the flood and dry seasons was 39.20% from 2013 to 2015 and 35.40% from 2016 to 2018. The topographic position has a certain effect on the TP and

TN concentrations, but human activities on topographic positions, such as the combined land use of topographic position, may have additional explanatory power on the TP and TN concentrations.

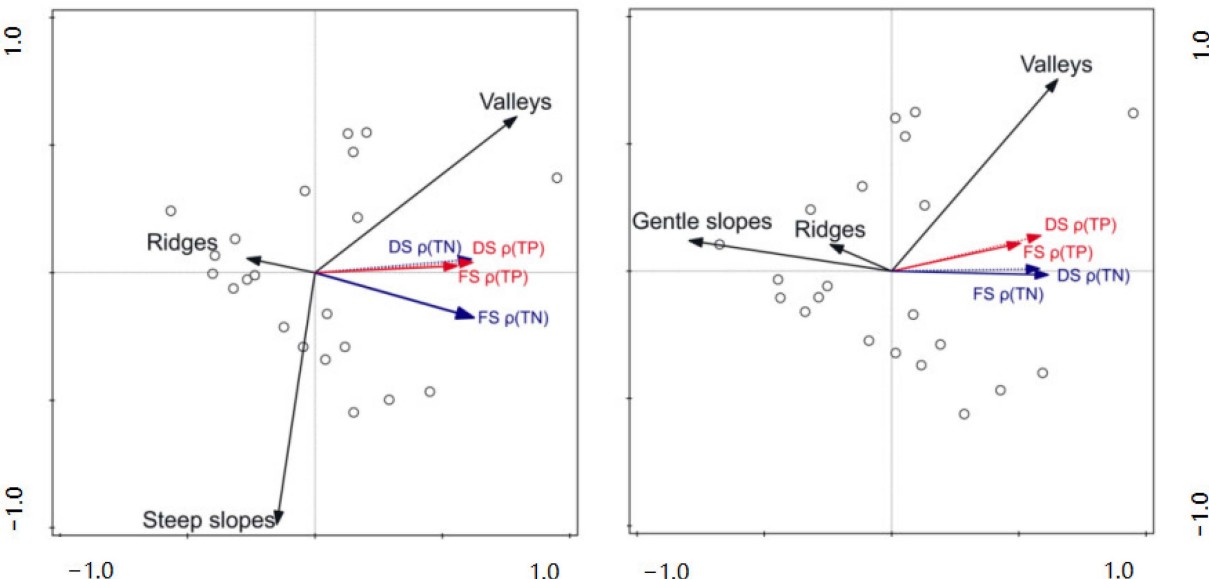

**Figure 16.** Redundancy analysis of TPI and TP and TN concentrations.

**Table 6.** Explanation of TP and TN by TPI (%).

| Period | Valleys | Gentle Slopes | Steep Slopes | Ridges |
|---|---|---|---|---|
| 2013–2015 | 24.3 | - | 14.6 | 0.3 |
| 2016–2018 | 1.3 | 22.4 | - | 11.7 |

To summarize, the correlations between the slope areas of different topographies and TN and TP decreased in the following order: valley (0.686) > gentle slope (−0.648) > steep slope (0.574). Due to the significant aggregation effect of valleys on TN and TP, the impact of topographic position on TN and TP should be comprehensively analyzed to distinguish the source–sink effects of different topographies in the basin. The control zoning can then be appropriately differentiated to reduce nutrient diffusion through the basin.

*3.5. Relationships between "L-SP" and Water Quality*

3.5.1. Correlation Analysis between "L-SP" and Water Quality

Figure 17 displays the Spearman coefficient analyses between the area proportions of composite land use types at the topographic slope sites and the TP and TN concentrations in the 21 monitoring units of the HBA watershed. The land use types were forest land, dry farmland, paddy land, grassland and construction land, and the topographic slope sites were valley, gentle slope, steep slope, and ridge. Since the distributions of paddy land, grassland and construction land are small on ridges, their correlations with ridges are not discussed. The land use types corresponding to the different slope positions were superimposed to obtain Figure 18.

During 2016–2018, positive correlations were reported between forest land in valleys and TP (0.473 and 0.598 in the flood and dry seasons, respectively) and between forest land in valley and TN in the flood season (0.472). From 2013 to 2015, the forest land on gentle slopes was correlated with TP in the flood and dry seasons (−0.749 and −0.678, respectively) and with TN (−0.809 and −0.693, respectively). The forest land on steep slope was correlated with TP in the flood and dry seasons (−0.702 and −0.661, respectively) and with TN (−0.64 and −0.681, respectively). From 2016 to 2018, the correlation between

TN and forest land on steep slopes was −0.508 in the flood season and −0.512 in the dry seasons. The correlations between ridge forest land and TP in the flood and dry seasons were −0.788 and −0.575, respectively, in 2013–2015 and −0.682 and −0.753, respectively, in 2016–2018. Moreover, the correlations between ridge forest land and TN in the flood and dry seasons were −0.815 and −0.740, respectively, in 2013–2015, and −0.788 and −0.797, respectively, in 2016–2018. All of these negative correlations are highly significant. The most stable correlation was that between TP and TN. The convergence of TP and TN was prominent in valley forest. In ridge forest, steep forest and gentle-slope forest, purification has been promoted by intercepting and blocking of TP and TN. The purification intensity was highest on ridge forest land, intermediate on gentle-slope forest land, and lowest on steep-slope forest land.

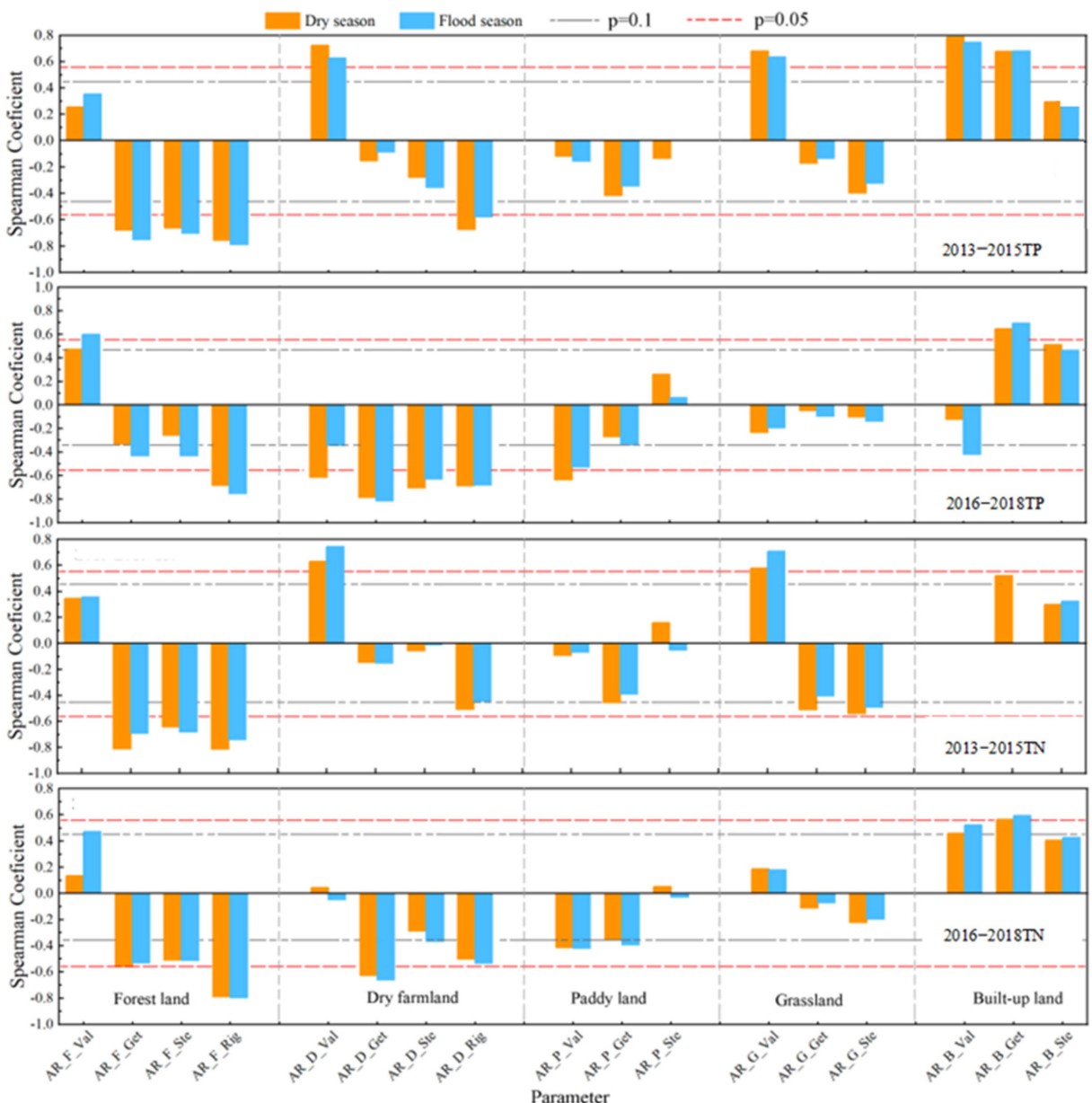

**Figure 17.** Correlations between land use structures and TP and TN concentrations on different topographic slopes.

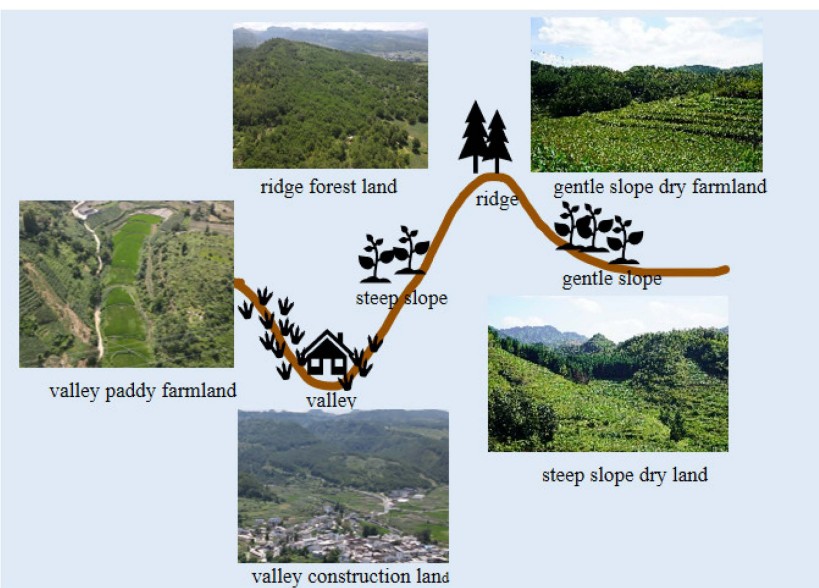

**Figure 18.** Different "L-SP" schematic diagram.

From 2013 to 2015, the correlations between valley dry farmland and TP in the flood and dry seasons were 0.627 and 0.722, respectively, and those between valley dry farmland and TN in the flood and dry seasons were 0.628 and 0.744, respectively. The TP and TN correlations were both significant. From 2016 to 2018, valley dry farmland was significantly correlated with TP ($-0.616$) during the flood season, whereas dry farmland was highly correlated with TP during the flood and dry seasons ($-0.785$ and $-0.815$, respectively) and significantly correlated with TN during the flood and dry seasons ($-0.626$ and ($-0.662$, respectively)). No obvious relationship between TP and TN was observed on steep-slope dry farmland. Over both periods, the correlation coefficients between ridge dry farmland and TP and TN were significant in the flood and dry seasons. The Spearman coefficients show that when the TP and TN concentrations are high, the valleys play a dominant role and the pollutant-convergence effect of dry farmland in the valley is prominent. In the different sloped dry farmland management methods, gentle slope land can take soil and water conservation tillage measures (e.g., conservation tillage or covering measures). With the increase in the slope, slope terrace, contour shrub-grass belt (hedgerow), and other short slope engineering measures can be adopted. When the slope increases further, terraced fields are generally used. Steep slope according to the "law of soil and water conservation "needs to return farmland to forest, construct slope soil, and conserve water in the forest.

Among relationships between paddy land on different topographic positions and the TP and TN concentrations, only valley paddy land was significantly related to TP ($-0.635$ and ($-0.528$ in the flood and dry seasons, respectively) from 2016 to 2018, and only gently sloped paddy land was related to TN ($-0.455$) during the dry season of 2013–2015. No obvious relationship appeared between topographic-slope compound paddy land and TP and TN concentrations, possibly due to the fact that paddy land is primarily distributed in valley areas, the distribution areas of gentle slopes, steep slopes and ridges are small, and planting activities are strongly dominated by human beings.

From 2013 to 2015, grassland was positively correlated with TP (0.635 and 0.680 in the flood and dry seasons, respectively) and TN (0.578 and 0.707 in the flood and dry seasons, respectively). From 2016 to 2018, gentle-slope grassland was negatively associated with TP ($-0.508$) in the flood season and steep-slope grassland was negatively associated with TP in both the flood and dry seasons ($-0.538$ and ($-0.490$, respectively). On other complex grasslands at different topographic positions, the correlations with TP and TN were mostly negative but insignificant. Compound grasslands with different topographic

positions are affected by ecological-land use and exert a weak purification effect on TP and TN concentrations.

From 2013 to 2015, highly significant correlations appeared between valley construction land and TP in both flood and dry seasons (0.745 and 0.785, respectively) and TN (0.855 and 0.785, respectively). Construction land on gentle slopes during the flood and dry seasons was positively correlated with TP (0.679 and 0.675, respectively, in 2013–2015 and 0.645 and 0.694, respectively, in 2016–2018) and with TN (0.521 and 0.534, respectively, in 2013–2015 and 0.564 and 0.593, respectively, in 2016–2018). From 2016 to 2018, steep-slope construction land was positively correlated with TP in the flood and dry seasons (0.510 and 0.463, respectively). On different topographic positions of each land use type, composite construction land was positively associated with TN and TP concentrations, reflecting the prominent characteristics of urban use. Therefore, construction land is an occurrence place of high TP and TN concentrations and must be targeted in pollution control strategies.

In general, for different slope land management, infiltration should be increased and runoff should be reduced in the valley. Soil and water conservation tillage measures can be adopted on gentle slope (e.g., contour strip tillage). With the increase in slope, we can adopt contour shrub grass band (hedgerow) and other short slope engineering. Steep slope needs to return farmland to forest, the construction of slope soil and water conservation forest.

3.5.2. Redundancy Analysis of Topographic-Slope Compound Land Use Types and TP and TN Concentration

In the redundancy analysis of area proportion of each land use type on TP and TN concentrations (Figure 14), the primary explainers of water quality were forest land, dry farmland, and construction land. Moreover, when compound land use types on different topographic slopes were correlated against TP and TN concentration, the topographic position obviously affected water quality. In the analysis of this subsection, the contribution rates of different topographic positions are combined with three land use types (Figure 19, Table 7). Since the distribution of ridge construction land was extremely small, it is not discussed here. Analyzing the land uses and TP and TN redundancies of topographic slopes from 2013 to 2015 and from 2016 to 2018, the interpretation rate of TP and TN was highest (50.90%) on gentle-slope forest land, followed by steep-slope forest land. In the dry farmland category, the high pollution output from 2013 to 2015 was primarily explained by TP and TN in valley dry farmland (explanation rate = 79.12%). When the nitrogen and phosphorus output decreased from 2016 to 2018, the explanation rate of gently sloping dry farmland to water quality was 22.11%. Among the construction land types, valley construction land most obviously increased the TP and TN concentrations, yielding the highest interpretation rate (88.70%).

To summarize, on the topographic slope scale, the land use type reflects human activities and the composite land use of topographic slopes makes higher contributions to TP and TN pollution than the topographic slopes themselves. Valleys occupy the lower part of the basin that collects the pollutants from steep and gentle slopes and upper and middle reaches of the basin. Therefore, valleys act as pollution sinks. When the pollution concentration is large, the cumulative interpretation rate of land use type is obviously higher for valleys than for other topographic slopes. On steep slopes and ridges, where the effects of human activities are small, the cumulative interpretation rates of different land use types on TP and TN are low. Pollution prevention and control can be achieved by comprehensively analyzing the effect of the topographic position of compound land use on nutrient migration through the watershed. In addition to analyzing the influence of land use and topographic slope position on TP and TN pollution, the source control area and pollution purification area in the basin must be systematically identified.

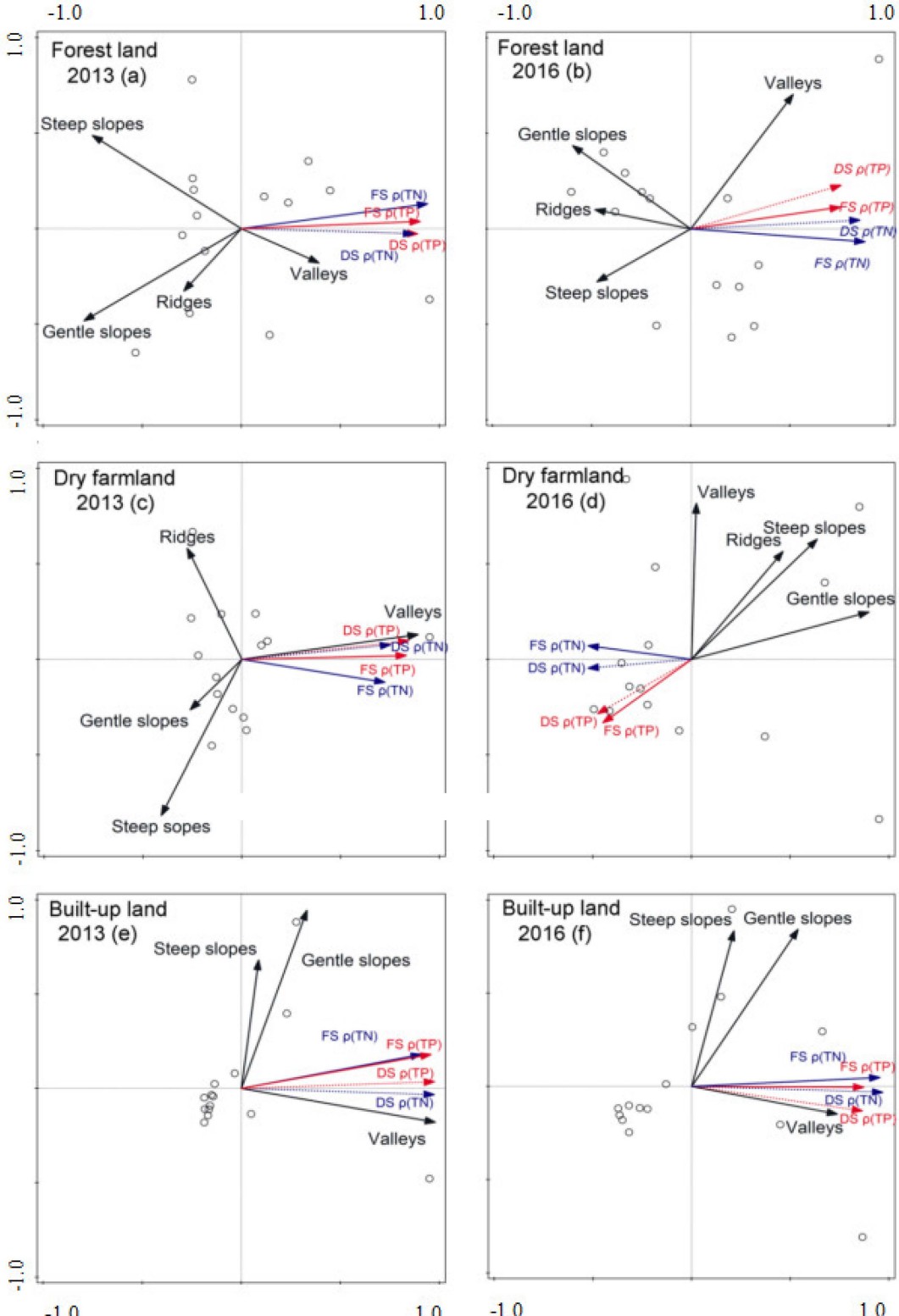

**Figure 19.** Redundancy analysis of land use composition and TP and TN concentrations in different TPI: Forest land in 2013 (**a**) and 2016 (**b**); Dry farmland in 2013 (**c**) and 2016 (**d**); Built-up land in 2013 (**e**) and 2016 (**f**).

**Table 7.** Interpretations of TP and TN by land use composition on different TPIs.

| Period | | Valleys | Gentle Slopes | Steep Slopes | Ridges |
|---|---|---|---|---|---|
| 2013–2015 | Forest land | 9.7 | 50.9 | 19.0 | 1.3 |
| | Dry farmland | 79.1 | 16.8 | 6.8 | 7.7 |
| | Built-up land | 88.7 | 3.6 | 1.0 | - |
| 2016–2018 | Forest land | 4.0 | 25.5 | 14.4 | 29.9 |
| | Dry farmland | 1.8 | 22.1 | 3.2 | 0.6 |
| | Built-up land | 50.3 | 12.5 | 29.7 | - |

## 4. Suggestions

The TP and TN concentrations were reported to depend on the slope of the terrain. The land use types of the HBA watershed can be divided into nitrogen and phosphorus purification areas (ecological land comprising forest land, grassland, and water bodies), nitrogen and phosphorus pollution areas (agricultural land comprising dry farmland and paddy land), and nitrogen and phosphorus pollution areas (urban land; also called construction land). As per their functions, the topographic slopes can be divided into source-prevention areas (ridges and steep slopes), sink-control areas (valleys), and control areas with no obvious pollution source (gentle slopes). The land use type was superimposed on the topographic position to correct the correlations between composite land use types at different slope positions and the TP and TN concentrations. Urban land, on which the output concentration of TP and TN is high, should be completely controlled and classified as a control area. Finally, the HBA watershed can be divided into ecological-land purification and prevention areas, ecological-land purification and control areas, agricultural land pollution control areas, agricultural land pollution control areas, and urban land pollution control areas. Figures 20 and 21 show the principles and results of this classification, respectively. The zoning results agree with the land use structure, spatial distribution pattern of topographic slopes, and land use degree of the HBA watershed, thus providing scientific reference value.

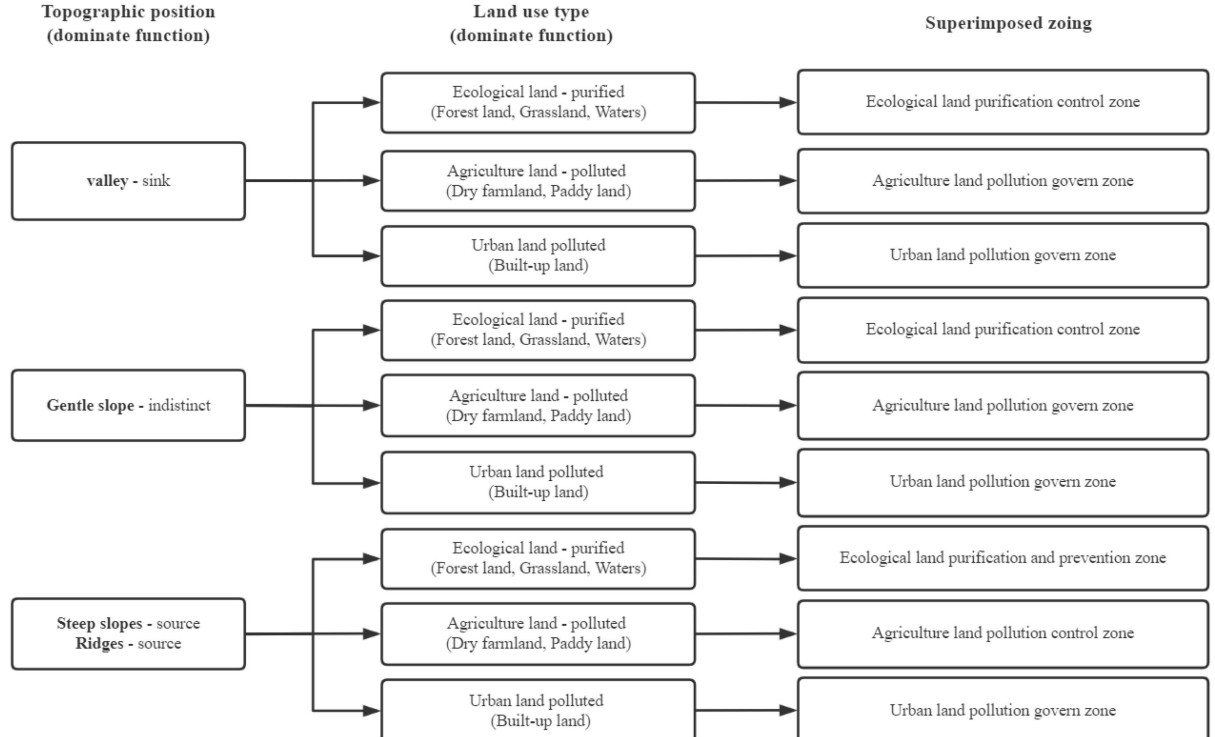

**Figure 20.** Principles of superimposed zoning.

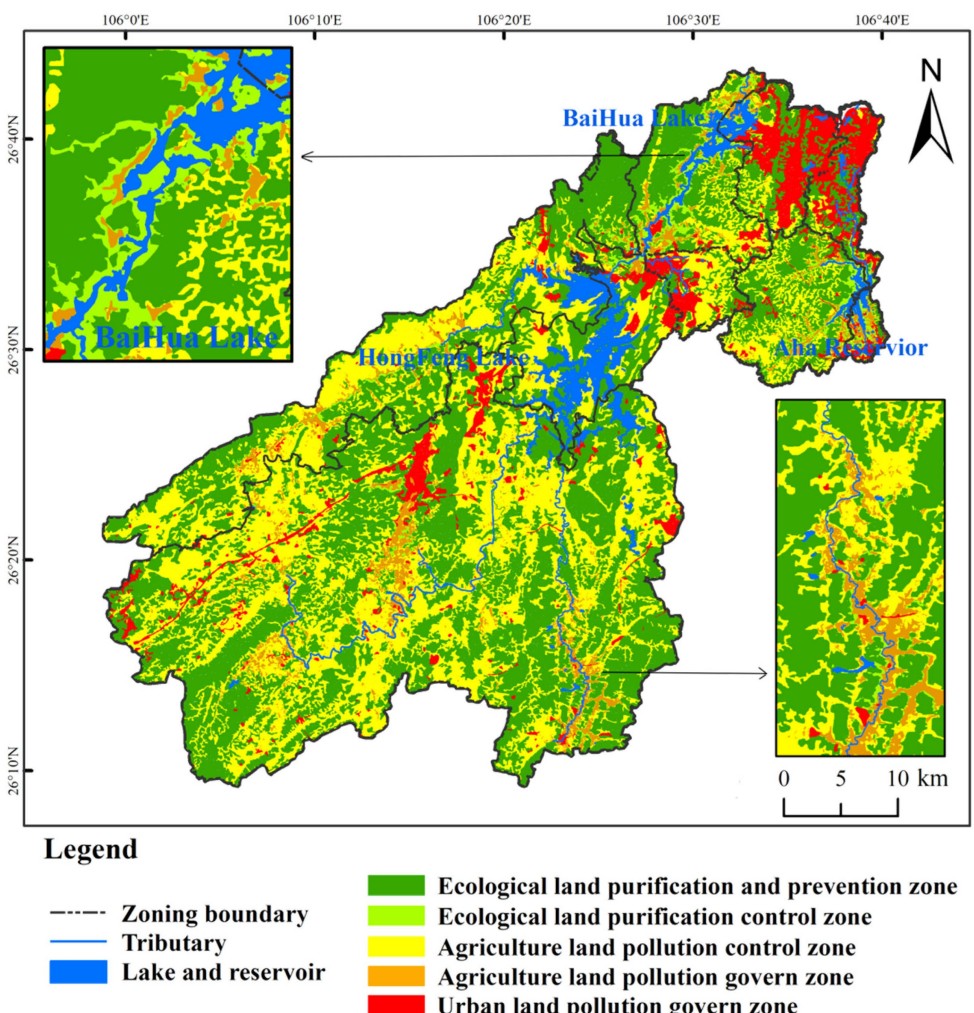

**Figure 21.** Key pollution control zones in the study area.

Ecological-land purification and prevention areas occupy 48.64% of the whole basin of the HBA watershed. The agricultural land, urban land, and ecological-land pollution control areas cover 25.48%, 7.97%, and 3.77% of the basin, respectively, and agricultural-land pollution governing zone occupy an area proportion of 4.31%. The ecological-land purification prevention areas and agricultural land pollution control areas are distributed over the whole basin, whereas the urban land pollution control areas are concentrated in the Baihua Lake and Aha Reservoir Basin. The agricultural land pollution control areas and ecological-land purification control areas are primarily distributed in the Hongfeng Lake Basin and Baihua Lake Basin, respectively.

Analyzing the different zones, the ecological-land purification prevention areas are characterized by improved ecological environment background and higher ecological service function than other areas. Therefore, they are the ecological safety barriers of the river basin and must be properly protected from development. Ecological-land purification control areas are mostly located in the transition zone between ecological land and agricultural land or along the banks of tributaries. They perform a certain purification function and are vulnerable to human activities. Therefore, human interference should be alleviated in these areas. Agricultural land pollution control areas are potentially polluted with nitrogen and phosphorus. Agricultural land is mostly distributed along the banks of tributaries, which carry a large number of non-point source pollutants that migrate and accumulate along the river. As tributaries have considerably higher nutrient concentrations than reservoir areas, they are the primary nutrient sources of reservoirs. To reduce these

inputs and lower the transport of nutrients from tributary sources, the use of chemical fertilizers and pesticides containing nitrogen and phosphorus in agricultural land pollution control areas should be limited. Agricultural control is extensively implemented since nitrogen and phosphorus pollution is easily transported along rivers and easily gathers to excessive levels in low-lying areas. Therefore, agricultural development activities should focus on minimizing the use of nitrogen phosphorus fertilizers and pesticides, cutting off the branch of pollutant source to the reservoir area, adopting biological purification technology, improving the tributary-farming pattern, reducing the effects of nitrogen and phosphorus, and improving the current situation of nitrogen and phosphorus storage. Urban land pollution control should target the output areas where nitrogen and phosphorus accumulate, since nutrients and town life pollutant emissions are carried downriver and increase the nutrient levels in the basin. Therefore, the area should be protected from the high pollution by urban development and downstream accumulation of nutrients. Thus, ecological protection schemes should be developed and the effect of urban sewage on lakes and reservoirs should be reduced.

## 5. Conclusions

In this study, the layout of the studied monitoring points comprehensively considered the natural and social factors in the basin to better reflect the eutrophication and spatiotemporal variation characteristics of TN and TP on annual and monthly scales. To improve the accuracy of land use data, we tested the classification accuracy using the Kappa coefficient. This study adopted the Carlson index method modified by a comprehensive nutritional status index. The correlations among the regional water quality parameters and the relationship between land use and water quality were evaluated through the Spearman correlation analyses. Using the same method, Lin et al. [37] and Liu et al. [38] successfully evaluated the eutrophication degree in the HBA watershed. Furthermore, Huang et al. [39] and Bo et al. [40] quantitatively explored the spatial and temporal relationships between land use and water quality parameters through a redundancy analysis, thus, obtaining the amounts of land use that explained different parameters in the HBA watershed. In future investigations, it will be necessary to explore the impact of climate change on water quality and ecology indices [41–43].

The presently obtained spatial and temporal change characteristics of eutrophication in the HBA watershed are consistent with those of Zeng et al. [44] and He et al. [45]. The primary conclusions are summarized below.

(1) Between 2013 and 2018, the primary water bodies in the HBA watershed were in a medium trophic state; The TLI of the interannual nutrient index demonstrated a weakly decreasing–increasing trend over the study period. The TLI was higher in the flood season than in the dry season and received an annually decreasing contribution rate from TN with no change in contribution rate of TP. This result shows a positive effect of TN control measures but no obvious effect of TP control measures.

(2) The ratio of TN to TP in Hongfeng Lake is the highest, where the effect of phosphorus is the strongest, while the lowest ratio of TN to TP is in the Aha Reservoir. The water quality in Sancha River basin, Huaqiao Basin, and the east of the Aha Reservoir area of Hongfeng Lake should be monitored.

(3) Over annual and seasonal monitoring periods, the TLI values and TN concentrations at each monitoring point of the HBA watershed decreased in the order of flood season > annual > dry season, whereas the TP concentration decreased in the order of dry season > annual > flood season.

(4) In the different monitoring units, the mean TLI, TP concentration, and TN concentration decreased in the order of Aha Reservoir > Baihua Lake > Hongfeng Lake. The TP concentration was significantly higher in the tributaries of Aha Reservoir and Baihua Lake than in the reservoir areas. At maximum, the TP concentration ratio between the tributaries and reservoir area reached 14.44. The tributaries are an important source of higher TP concentration in the reservoir areas. In Honfeng Lake, the TN concentration was

significantly higher in the tributaries than in the reservoir area but no obvious differences in TP concentration were observed. As the Aha Reservoir is located in the lower reaches of the basin, the migration and accumulation of nutrients along the river produce a more severe eutrophication trend in this reservoir than in Baihua Lake and Hongfeng Lake, and the situation becomes serious in the flood season. Construction land and cultivated land were identified as pollution sources while woodland is a pollution sink and valleys tend to aggregate TN and TP. Therefore, control and management zones should be determined after comprehensively analyzing the influences of topographic positions and land uses on the eutrophication of the basin.

(5) The area proportions of different land use types were correlated with the TN and TP concentrations in the HBA watershed. The TN/TP concentrations were significantly positively and negatively correlated with construction land and dry farmland, respectively (with maximum coefficients of 0.698 and −0.662, respectively). Forest land exerted a significant purification effect on TN and TP while construction land was a pollution source. Areas of different slope topographies were correlated with TN and TP as follows: valley (0.686) > gentle slope (−0.648) > steep slope (0.574). Valleys tended to accumulate TN and TP. The area proportions of different topographic slopes combined with different land use types were correlated with TN and TP as follows: valley construction land (0.855) > gentle-slope dry farmland (−0.815) > ridge forest land (0.814) > valley grassland (0.707) > valley paddy land (−0.635). Both the composite conditions of land use and topographic positions should be comprehensively considered for the prevention and control of TN and TP zoning.

(6) The pollution degree of different land uses decreased in the order of urban land use > agricultural land > ecological land. There are different urgency levels of governance (i.e., governance area > control area > prevention area). The comprehensive land uses and topographic slopes can be divided into ecological-land purification protection zones, ecological-land purification control zones, agricultural land pollution control zones, agricultural land pollution control zones, and urban land pollution control zones. Different sources of nutrients were identified in the tributaries and reservoir areas of the HBA watershed. A differential management scheme has reduced the nitrogen and phosphorus nutrients in the tributaries and blocked their entry to the reservoir. Reduction measures for non-point source pollution dominated by TP and TN should be optimized via source reduction, interception, and remediation. The pollutant source should be reduced via cultivation and nutrient management, the interception of riverbank buffer zones and ecological wetlands, and ecological restoration projects such as returning paddy land to forests or grasslands.

**Author Contributions:** Conceptualization, X.J. and Y.P.; Methodology, X.J.; Software, X.J. and Y.P.; Validation, X.J., Y.P. and W.Z.; Formal Analysis, X.J., Y.P. and W.Z.; Investigation, X.J.; Data Curation, Y.P. and W.Z.; Writing—Original Draft Preparation, X.J.; Writing—Review & Editing, Y.P., X.Z. and W.Z.; Visualization, Y.P. and W.Z.; Supervision, X.Z.; Project Administration, X.Z.; Funding Acquisition, X.Z. and S.Y.; Resources, S.Y. All authors have read and agreed to the published version of the manuscript.

**Funding:** This research was funded by The National Natural Science Foundation of China. Further, the project was jointly funded by the Karst Science Research Center of Guizhou Provincial People's government: the calcium-dependent mechanism of biodiversity formation and maintenance in Karst and its application basis (U1812401); Guizhou Province Science and Technology Support Project ([2017]2855); Basic Research Program of Guizhou Province ([2017]1131).

**Institutional Review Board Statement:** Not applicable.

**Informed Consent Statement:** Not applicable.

**Data Availability Statement:** The water data in Hongfeng Lake and Baihua Lake was sampled on a monthly basis from 2013 to 2018. The water data in the Aha Reservoir was sampled at three-monthly intervals from 2013 to 2014 and on a monthly basis from 2015 to 2018. The digital elevation model data that support the findings of this study are openly available in [the Geospatial Data Cloud] at [http://www.gscloud.cn/], accessed on 13 January 2019. The land use data that support the

findings of this study are openly available in [the website of the China Resources Satellite Data and Application Center] at [http://www.cresda.com/], accessed on 10 January 2019.

**Acknowledgments:** The author express their gratitude to Luo Ya, Zhou Qiuwen and Yang Guangbin from Guizhou Normal University for their guidance in the field research work. Some of the basic data are provided by Gao Chengcheng and Huang Xueyong of Guiyang Institute of Water Resources and Hydropower Survey and design.

**Conflicts of Interest:** The authors declare no conflict of interest. The sponsors had no role in the design, execution, interpretation, or writing of the study.

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
