# Peer review of "Zoning Strategy for Basin Land Use Optimization for Reducing Nitrogen and Phosphorus Pollution in Guizhou Karst Watershed"

_water, doi:10.3390/w14162589_

Round 1
Reviewer 1 Report
General comments:
The submitted manuscript represents a zoning strategy for optimization of basin land use to reduce nutrient pollution in Guizhou karst watershed. The manuscript is quite interesting, and the findings could be helpful for watershed managers in developing strategies for sustainable watershed management. However, a few issues still need to be addressed before further processing the submitted article.
Specific comments:
1. Mention the objectives of the study in the abstract section.
2. L16: ‘HBA’- write in complete form.
3. L29: Consider deleting the phrase 'because of agricultural pollution'.
4. L31: Replace the word 'pollution' with either 'activities' or 'practices'.
5. L34: Better to use the words like 'leaching' rather than ‘underground leakage’.
6. L38: Remove the word 'human'.
7. L47: Replace the word 'emissions' with 'loading'.
8. L79: ‘TLI’- explain at the time of first use.
9. Section 2.1: It would be better to represent the study area characteristics in tabular form.
10. How about the amount of annual precipitation?
11. Figure 1: There are 27 monitoring units indicated in the figure.
12. L163: The authors mentioned the total number of monitoring units was 21, while in L143, they wrote ‘28 monitoring points were arranged’. Provide clarification.
13. L205: ‘With’- not clear.
14. Section 2.3.4: Add some description of the technical route.
15. The results section is, in general, fine but very long. On the other hand, the discussion section is not well developed. It would be better to merge the discussion section with the results. Another suggestion is the authors can highlight the ‘Suggestions’ as a separate section.
16. The language of the manuscript requires some improvement, especially in terms of wording.
Author Response
List of Responses
Dear Editors and Reviewers:
Thank you for your letter and for the reviewers’ comments concerning our manuscript entitled”Zoning strategy for basin land use optimization for reducing nitrogen and phosphorus pollution in Guizhou karst watershed”. Those comments are all valuable and very helpful for revising and improving our paper, as well as the important guiding significance to our researches. We have studies comments carefully and have made correction which we hope meet with approval. Revised portion are marked in red in paper. The main corrections in the paper and the responds to the reviewers’ comments are as follows:
Reviewer #2:
- Response to comment:(It should be” posing a serious threat”, not “posing serious threat”.)
Response:We have re-written this part according to the Reviewer’s suggestion.
- Response to comment:( It might be better to introduce food production before all life on Earth and on line 38 – not sure what is meant by human aquatic ecosystems )
Response:We have changed the word order of the original text, putting food production ahead of all life on Earth. Also, with regard to the latter question, I think it is due to our translation error that we have deleted the word “human”.
- Response to comment:(Ln 48 – not sure what dredging bottom mud of Lily Lake means )
Response: Dredging means removing silt or digging deep to allow water to flow, and in the original article refers to some of the work done in the United States to clean up water pollution in river basins.
- Response to comment:(It is not clear to me what the authors are referring to when they say that nutrient concentrations are higher in non-point pollution sources than in lakes & reservoirs.)
Response: What this means is that the tributaries take in more sewage, and in general, the nutrient concentration in the tributaries is usually higher than in the reservoir area, so we are in the process of managing the lake basin, the management of tributaries should be strengthened.
- Response to comment:(What is meant by “compound land-use” is it referring to compounds with multiple dwellings or are the authors just referring to land-us whether it be agriculture, natural forest or urban development?)
Response: In this paper, we redefine the term“L-SP”, which not only refers to different types of land use, but also takes into account the different land-use patterns on different slope position in order to deepen the study.
- Response to comment:(TLI should be spelled out in the text. Where do the values presented come from)
Response:About the first one, we have re-written this part according to the Reviewer’s suggestion. About the second one, we refer to the research made by shengxing long et al, and the TLI value of the study area comes from Guiyang environmental monitoring station and “two lakes and one reservoir” management monitoring center station.
- Response to comment:(Authors might consider using “objectives” instead of “purpose”)
Response: we have re-written this part according to the Reviewer’s suggestion.
- Response to comment:(Actual rainfall amounts for the year orfor the wet and dry seasons should be provided.)
Response: “The annual average rainfall was 1129.5 mm, of which 593 mm in dry season and 1386 in wet season” was added.
- Response to comment:(What is meant by the water quality being basically stable in Category II-III – what are thecategories)
Response: According to the”Surface water environmental quality standard”(GB3838-2002) , China's surface water is divided into five categories. Class II is mainly applicable to the first class protection area of the surface water source of centralized drinking water, rare aquatic habitat, fish and shrimp production, fish feeding grounds for young and young fish, etc. . Class III is mainly applicable to the secondary protection area of the surface water source of centralized drinking water, the wintering grounds for fish and shrimp, migratory channels, aquaculture areas and other fishery waters and swimming areas. In 1980, Guiyang's water quality could be classified into Class II.
- Response to comment:(The first question is whether the coal mining mentioned in this article is open-pit mining or underground mining? The second suggestion is that the colored dots in Figure 1 are indistinguishable and that it is not clear where the sub-basin monitoring units are and what the numbers in the figure represent.)
Response: First, to answer the first question, the coal mining mentioned in this paper is underground mining. Second, we have reworked figure 1. In the picture, the green dots indicate the monitoring points of tributaries and the red dots indicate the monitoring of the reservoir area. And the numbers in the graph represent different monitoring units. The contour line of the sub-basin monitoring unit is a gray dotted line without shadow. In addition, figure B is a local magnification of Figure A, in order to better distinguish the monitoring points of tributaries and reservoir areas, so they are made to distinguish.
- Response to comment:(Where the different monitoring points are in the watersheds fromFigure 1. AndAnd there are only 27 points, not 28.)
Response: We have reworked figure 1 and noted the names of the different watersheds. Now we can count 28 points in the figure.
- Response to comment:(what is meant by “area of prospective drinking water source” for the two lakes would that not include the whole watershed of the streams flowing into the lakes.)
Response:We are sorry for the misrepresentation in this article. What we mean isThe first- and second-level protection zones and quasi-protection zones of drinking water source of HBA watershed
- Response to comment:(The number of points in Figure 1 does not correspond to the number of points in the paper. Is this the number of monitoring units?)
Response:We have reworked figure 1. Now we can count 28 points in the figure.And the numbers in the graph represent different monitoring units which are not equivalent to monitoring points.
- Response to comment:(The authors identified 400 different land-use types or 400 areas that could be classified into their 7 listed categories? It is not clear in that list what is meant by unused land – could that not include dry land and woodland and grassland?)
Response:I'm sorry that we misstated the content. What we really wanted to say is that we have collected 400 land use verification points on the ground. According to the current classification of land use types in China, we classified the 400 sites into seven land use types. The answer to the latter question is that the present division of land in our country takes landform, soil and vegetation as the main marks, and unutilized land usually means that there is no vegetation cover and has not been utilized by human beings, temporarily undeveloped land.
- Response to comment:(Why would dry farmland be classified as agricultural land? Also why even mention construction land if it means urban land?)
Response:First, to answer the first question, according to the national classification of land use status, drylands refer to land without irrigation facilities, where dryland crops are grown mainly on natural precipitation, and are therefore classified as agricultural land. Secondly, For the second question, the construction land mentioned in the article mainly refers to buildings, while urban land also includes land other than buildings, such as green space, squares, etc. , so we also classify construction land as urban land.
- Response to comment:(Why not just start with table 2 and skip much of the previous descriptions to shorten this very long paper? Paddy land, unused land and built-up land could be introduced earlier in the text.)
Response: For this comment, we have deleted the original Table 2. Because this section is the source and classification of land use data, not the core part of the text, so we do not have a description of each type of land.
- Response to comment:( What is meant by low, medium and high slope position – are these the positions a slope with low being the footslope, medium the backslope and high the shoulder and summit?)
Response: We took a look at Lay's article and replaced it with“Uphill, mid-slope, and downhill,” which refers to the upper part of the slope that is triangulated from the ridge down to the valley, the middle slope is the middle part, and the downhill slope is the lower part.
- Response to comment:(What waterland grade actually means in terms of land use does it refer to waterquality? Also not sure what agricultural land level means in terms of land use and what is an artificial grassland – is this a created grassland where forest may have been converted to grassland? And why is mining land included under urban land – that would suggest to me that there are mines within city limits?)
Response: First, we have redefined table 4, which may be more scientific and reasonable. Second, artificial grassland refers to standard hay, grass grass harvested from booting to heading and legume grass harvested from early flowering. Finally, according to the“Land Management Law”, the mining land belongs to the construction land. And we have defined the mining land as non-renewable land category.
- Response to comment:(The length and direction of the arrows is unclear in Ln 254.what the different zones are?how does that differ from the ecological land purificationcontrol zone and what are the differences between the two agriculture zones, etc. )
Response: First, we have graphs for the rest of the article, as shown in Figure 13. In this section, only the redundancy analysis method is introduced. To the latter question, we have answered it in the conclusion part of the article. Here we are just explaining the technical route.
- Response to comment:(What kind of TP control practices have been introduced in the watersheds where TP has been reduced?)
Response: Considering the Reviewer’s suggestion,we have added the following to the corresponding section, “The main method is to explore the eco-agricultural model of efficient utilization of soil nutrient elements by using ecological principles, so as to make farmland harvest a variety of high-quality agricultural products. At the same time, green manure planting, vegetable fertilizer dual-use, organic fertilizer instead of fertilizer methods to optimize the planting structure.”
- Response to comment:(What “the treatment” refers to?)
Response:“Treatment” refers to measures taken to reduce total nitrogen and total phosphorus. These measures refer to a administration bureau of HBA watershed was set uo to ban the production, sale and use of phosphorus-containing detergents within the river basin, and comprehensively strengthen the implementation of biological purification and restoration projects.
- Response to comment:( Is the water that Guiyang citizens drink treated before citizens drink it or do they drink water that is directly piped out of the rivers or lakes?)
Response: The water that Guiyang citizens drink treated before citizens drink it. HBA watershed as the main source of drinking water, the concentration of TN and TP will affect the water quality, so it is necessary to take measures to reduce TN and TP.
- Response to comment:(What is meant by the word paurophic? )
Response: Paurophic refers to poor nutrition, or lack of nutrients, as opposed to eutrophication.
- Response to comment:(Is the idea that phosphorus moves with the sediment, so during the dry season there are fewer floods, resulting in unmoved concentrations of phosphorus?)
Response:What we are trying to say is that phosphorus is a sedimentary substance. During the dry season, the concentration of phosphorus in the whole lake increases because of less flood events and the deposition of phosphorus on the bottom of the lake.
- Response to comment:(Explaining the potential reasons for the different concentrations of total phosphorus in different watersheds, as well as the specific land-use patterns and agricultural activities in different watersheds.)
Response:Considering the difference of land use in different river basins, Hongfeng Lake is located in the upper reaches of the river, playing a variety of roles such as drinking water source, power generation, tourism and so on. In addition, the land types on both sides of Hongfeng Lake basin are paddy fields. Rice is the main crop, with sporadic coal mines near the upper reachesBaihua Lake is located in the downstream of Hongfeng Lake, so it is more polluted than Hongfeng Lake, however, the problem of water quality is becoming more and more serious in Aha reservoir as sewage and garbage were discharged into the Aha reservoir and the peasants built their houses as a kind of farmhouse.
- Response to comment:(Describing more specific discussion about the difference in the movement of soluble N and sediment attached P in this discussion and explaining how that relates to developing methods to control their movement from the land to the water bodies.)
Response: The phosphorus load in the sediments of Hongfeng Lake is high. And when the climate falls, the vertical convection of the water body rises, which leads to the increase of phosphorus content in the water body. Baihua Lake and Aha reservoir are rainfall-recharging reservoirs, so during the dry season, phosphorus deposition leads to an increase in phosphorus concentration. For nitrogen, during the flood season, heavy rainfall increases surface runoff which leads to the loss of fertilizer and increases the nitrogen content.
- Response to comment:(In the recommendation, figure 4 can represent points with lower values in different colors.)
Response: Considering the Reviewer’s suggestion,we have redrawn figure 4.
- Response to comment:(It should be” is present at low levels”, not “presents at low levels”)
Response: After our careful examination, we found no problem with the expression of the original text, so we did not make any changes.
- Response to comment:(Basin-to-basin comparisons should be accounted for in terms of specific land use and agricultural activities)
Response:The analysis of nitrogen-to-phosphorus ratio was based on the spatial variation characteristics of total n and P concentrations, so the analysis of the correlation between this part and land use was roughly the same as the new content in 3.1.2. Hongfeng Lake is located in the upper reaches of the river, playing a variety of roles in many aspects. In addition, the land types on both sides of Hongfeng Lake basin are paddy fields. Rice is the main crop, with sporadic coal mines near the upper reachesBaihua Lake is located in the downstream of Hongfeng Lake, so it is more polluted than Hongfeng Lake, however, the problem of water quality is becoming more and more serious in Aha reservoir as sewage and garbage were discharged into the Aha reservoir and the peasants built their houses as a kind of farmhouse.
- Response to comment:( Why TN and TN concentrations are both in the title – both are compared based on differences in concentrations are they not? Should on of these be TP as both are discussed in this section.)
Response:We are sorry that we made a mistake in typing the word. And we have re-written this part according to the Reviewer’s suggestion.
- Response to comment:( The tributaries or rivers are usually the major source of nutrients in reservoirs and so management should be focused on land-use changes in the headwaters of the first and second order streams of the watersheds.)
Response:Thank you for agreeing with us. According to the results of the study, the tributaries are more direct to receive municipal and agricultural sewage, so the management of the tributaries and sources of land-use change should be strengthened.
- Response to comment:(The fact that the tributaries of Hongfeng Lake and Baihua Lake are not affected by too much industrial drainage is not the reason why the CONTN and CONTP values of the reservoirs are higher than those of the tributaries.)
Response: We have made some modifications. The revision is that there are most of the living sewage along the Hongfeng Lake, large-scale tourism. In addition, the industrial waste water around Baihua Lake and agricultural practices discharge the waste water directly into the reservoir area is the reason
- Response to comment:(It would be good for discussing whatforms as fertilizer and what methods would be used to reduce the arable land unsuitable for cultivation ?)
Response: In response to the comments of the reviewers, we considered the use of organic fertilizers instead of highly polluting fertilizers.In addition, the inappropriate cultivation mentioned in the article mainly refers to the cultivation method with large fertilizer application, so that the use of chemical fertilizer can be reduced, while promoting the cultivation of legumes
- Response to comment:(What are the ranges for each of the slope classes presented?)
Response:The topographic slope index was calculated by ArcGIS, in which the flat area with TPI value close to 0 and inclined trend was gentle slope, usually with a degree of 3-10 ° , and then the degree of steep slope was usually more than 45 ° . The higher the value of TPI shows positive growth is the ridge, showing negative growth is the valley.
- Response to comment:(It is not clear when it is stated that ridges are concentrated in relatively flat areas.)
Response:We are sorry for the misrepresentation that ridges should be distributed at higher elevations, not in relatively flat areas.
- Response to comment:(It would be interest to know why the various changes in land-use occurred not just that they)
Response: In the early period of economic development, land use was extensive and greatly affected by human activities. Then, under the influence of the policy of returning farmland to forests and reclaiming land, ecological projects such as Afforestation were implemented. These were the main reasons for the change in land use.
- Response to comment:( How the land is actually managed?&what specifically has been done to control chemical fertilizers and pesticides that has reduced TN and TP concentrations?)
Response: Considering the Reviewer’s suggestion, the specific methods of land management including banning farmhouses along lakes (reservoirs) , banning the use of chemical fertilizers in powerful agricultural activities, adjusting the structure of agricultural production, and developing modern and organic agriculture have been added.
- Response to comment:( How the water bodies tend to purify ecological land – while there may be plants and biological organisms that can reduce TP and TN in the water that has no direct effect on the land that generated the TP and TN runoff.)
Response: We pondered the question posed by the reviewer, and our answer to that question was that the discharge of municipal sewage and agricultural water into lakes and reservoirs would increase eutrophication, which will directly affect the habitats on which the vascular plant depend. So we can infer that if we protect the water bodies and take measures to treat the sewage, we can purify the ecological land to some extent.
- Response to comment:( What the actual management practices are on the different slopes of dry farmland?)
Response: For different sloped dry farmland management methods, gentle slope land can take soil and water conservation tillage measures, such as conservation tillage (covering measures) and so on. With the increase of slope, slope terrace, contour shrub-grass belt (hedgerow) and other short slope engineering measures can be adopted. When the slope increases further, terraced fields are generally used. Steep slope according to the“Law of soil and water conservation” needs to return farmland to forest, the construction of slope soil and water conservation forest.
- Response to comment:(It would be very helpful to have a detailed presentation of the actual land management practices on the different land slope positions)
Response: For different slope land management, the valley should increase infiltration and reduce runoff. Soil and water conservation tillage measures can be adopted on gentle slope, such as contour strip tillage. With the increase of slope, we can adopt contour shrub grass band (hedgerow) and other short slope engineering. Steep slope according to the“Law of soil and water conservation” needs to return farmland to forest, the construction of slope soil and water conservation forest.
- Response to comment:(What the pollution purification areas actually consist of in terms of vegetation and land-us? )
Response: We are grateful to the reviewer for his/her comments. In this section, we want to express that comprehensive analysis of land use complex slope positionon the impact of nutrients can be scientific identification of watershed source control area, pollution purification area. What constitutes a pollution purification area is described in the final conclusion.
- Response to comment:(What an ecological land purification control zone should be? )
Response: Most of the ecological land purification control areas are located on the transition zone between ecological land and agricultural land, or along the tributaries, which have certain purification functions and are easily damaged by human activities, the disturbance of human activities should be mitigated. According to the principle of superposition division in figure. 19, the gentle slope forest land, gentle slope grassland and gentle slope water were classified as the ecological land purification control zone.
Special thanks to you for your good comments.
We tried our best to improve the manuscript and made some changes in the manuscript. These changes will not influence the content and framework of the paper. And here we did not list the changes but marked in red in revised paper.
We appreciate for Editors/Reviewers’ warm work earnestly, and hope that the correction will meet with approval. Once again, thank you very much for your comments and suggestions

Round 2
Reviewer 1 Report
The reviewer appreciates the authors' efforts to improve the quality of the manuscript. After the revision, the manuscript as a whole is more logically described and scientifically sound.
Author Response
Dear Editors and Reviewers:
Thank you again for your letter and for your kind work on the process of dealing with our manuscript. We thank you very much for giving us an opportunity to revise our manuscript, we appreciate editors and reviewers very much for your positive and constructive comments and suggestions on our manuscript entitled”Zoning strategy for basin land use optimization for reducing nitrogen and phosphorus pollution in Guizhou karst watershed”(ID: water - 1819620)
Thank you and best regards.
Your sincerely,
Wenbin zhang
11.Aug., 2022

Reviewer 2 Report
Thank you for considering the modifications that were suggested – the paper has been dramatically improved. A quick review of grammar should still be done.
Line 71 “…slope positions..”
Ln 128 – it would be helpful to know whether Category II is at the better end of the standard or the poorer end.
Table 4 is much clearer now – thank you.
Line 264 – Technical Route - Thank you for the description of the methodology that was used – it is now clear to me what was done and in what order it was done.
Lines 349-353 – valuable addition to the discussion
Lines 399-407 – this added explanation helps the reader a lot in understanding the land-use and their potential impacts on water quality in the lake.
Line 406 – not quite sure of the value or what the phrase “and peasants built their houses as a kind of farmhouse” – what impact do the different style of houses have on TN or TP or runoff?
Line 500 – “Large scale tourism occurs in most of the living sewage along ..” This does not make any sense to me?
Line 532 – a helpful addition at the beginning of this section also good to repeat in line 589.
Line 628 – helpful addition but not clear what is meant by “banning farm-house activities…” does that mean no houses can be built or that specific activities in the “yards” of houses along the lake are outlawed?
Line 795 – good summary addition to this section.
I like the addition of Figure 17 – it would nice if the pictures or the whole figure could be a little larger
Line 901 – good addition to the conclusion.
A general comment – please have someone review the grammar of some of the great new additions to the paper to make sure that the grammar is correct.
Author Response
Dear Editors and Reviewers:
Thank you again for your letter and for your comments concerning our manuscript. I should like to express my appreciation to you and the referees for suggesting how to improve our paper. Revised portion are marked in red in paper. The main corrections in the paper and the responds to the reviewers’ comments are as follows:
- Comment:(Replace the word 'slope position' with 'slope positions' in line 71.)
Response: We have re-written this part according to the Reviewer’s suggestion.
- Comment:(Ln 128 – it would be helpful to know whether Category II is at the better end of the standard or the poorer end.)
Response: Considering the Reviewer’s suggestion, we have re-written this part. Now the content of this section is”According to the ‘Surface water environmental quality standard (GB3838-2002),’ surface waters in China are divided into five categories, of which one to five categories represent the water quality from the best to the worst. “
- Comment:(What the phrase “and peasants built their houses as a kind of farmhouse”in line 406 & What impact do the different style of houses have on TN or TP or runoff?)
Response: I am sorry that we may have made a mistake in translation and you may not be able to understand what we are trying to say in the part. What we are trying to say is that the farmers have turned their houses into a place of entertainment for tourists to enjoy, such as picnics or barbecues. Therefore, these activities will lead to more sewage discharge into lakes, and increase the eutrophication of rivers. And the part has been changed to “the peasants built their houses as a kind of entertainment for tourists.”
- Comment:(“Large scale tourism occurs in most of the living sewage along ..” whichdoes not make any sense to the revi)
Response: What we want to analyze is why the ConTN of reservoir in Hongfeng Lake is higher than that of the tributaries. One of the reasons is that there are some agricultural activities along the of Hongfeng Lake, where rice is the main crop and sporadic coal mines are found near the upper reaches, which lead to the generation of most of the living sewage along the lake. So we add this as an explanation.
- Comment:(Line 532 – a helpful addition at the beginning of this section also good to repeat in line 589.)
Response: According to the Reviewer’s suggestion, we have re-written this part in line 611-613.
- Comment:(Line 628 – helpful addition but not clear what is meant by “banning farm-house activities…” does that mean no houses can be built or that specific activities in the “yards” of houses along the lake are outlawed?)
Response: I am sorry that we may have made a mistake in translation and the response to this question is about the same as the response to the third question. What we are trying to say is that the farmers have hold agritainment, a kind of farm-based tourism, such as picnics or barbecues, which will lead to more sewage discharge into lakes and increase the eutrophication of rivers. Therefore, the restrictions on recreational activities that we have described in this article are aimed at activities that are highly polluting, not at activities that are outlawed or totally prohibited.
- Comment:(It would nice if the pictures or the whole figure could be a little larger.)
Response: According to the Reviewer’s suggestion, we have reworked the whole figure.
Special thanks to you for your good comments.
We tried our best to improve the manuscript and made some changes in the manuscript again. These changes will not influence the content and framework of the paper.
We appreciate for Editors/Reviewers’ warm work earnestly, and hope that the correction will meet with approval. Once again, thank you very much for your comments and suggestions.
Your sincerely,
Wenbin zhang
11.Aug., 2022
